# The functional organization of descending sensory-motor pathways in *Drosophila*

**Shigehiro Namiki[1], Michael H Dickinson[2], Allan M Wong[1], Wyatt Korff[1], Gwyneth M Card[1]\***

[1]Janelia Research Campus, Howard Hughes Medical Institute, Ashburn, United States; [2]Division of Biology and Bioengineering, California Institute of Technology, Pasadena, United States

**Abstract** In most animals, the brain controls the body via a set of descending neurons (DNs) that traverse the neck. DN activity activates, maintains or modulates locomotion and other behaviors. Individual DNs have been well-studied in species from insects to primates, but little is known about overall connectivity patterns across the DN population. We systematically investigated DN anatomy in *Drosophila melanogaster* and created over 100 transgenic lines targeting individual cell types. We identified roughly half of all *Drosophila* DNs and comprehensively map connectivity between sensory and motor neuropils in the brain and nerve cord, respectively. We find the nerve cord is a layered system of neuropils reflecting the fly's capability for two largely independent means of locomotion – walking and flight – using distinct sets of appendages. Our results reveal the basic functional map of descending pathways in flies and provide tools for systematic interrogation of neural circuits.
DOI: https://doi.org/10.7554/eLife.34272.001

## Introduction

The evolution of nervous systems is dominated by the process of cephalization, in which anterior ganglia fuse to create a brain that integrates information from a number of specialized sensory organs (*Bullock and Horridge, 1965*). In most animals, this large cephalic structure communicates with motor centers via a diverse population of descending neurons (DNs), with axons that run in connectives, or tracts, to more posterior ganglia. As the sole conduits of information from the brain to the posterior ganglia within the nerve cord, the DNs play a key role in behavior. Their activity initiates or modifies central pattern generators in the nerve cord, thereby controlling locomotion without necessarily conveying the details of motor actions (*Lemon, 2008*; *Heinrich, 2002*). Activation of some individual DNs is sufficient to elicit quite specific coordinated actions (*Kien and Altman, 1984*; *Nolen and Hoy, 1984*; *Kohatsu et al., 2011*; *von Philipsborn et al., 2011*; *Bidaye et al., 2014*; *von Reyn et al., 2014*).

Because the number of DNs is several orders of magnitude smaller than the number of neurons in either the brain or posterior ganglia, this class of cells represents a critical bottleneck in the flow of information from sensory systems to motor circuits. The DNs are thus a strategic target for investigating sensory-motor processing, and a wiring diagram of their connectivity is critical to understanding how the nervous system controls behavior (*Burrows, 1996*; *Drew et al., 2004*). Although previous studies have revealed the broad organization of descending pathways in a range of species (*Breidbach, 1990*; *Kanzaki et al., 1994*; *Staudacher, 1998*; *Okada et al., 2003*; *Cardona et al., 2009*; *Hsu and Bhandawat, 2016*; *Severina et al., 2016*; *Kien and Altman, 1984*; *Strausfeld et al., 1984*; *Strausfeld, 2012*; *Lemon, 2008*), uncovering the sensory-motor mapping represented by the DNs requires analysis of inputs and outputs with single-cell resolution. The genetic model organism, *Drosophila melanogaster*, offers an opportunity to pursue such a systematic analysis. Flies are

**\*For correspondence:**
cardg@janelia.hhmi.org

**Competing interests:** The authors declare that no competing interests exist.

capable of complex behaviors, yet possess a tractable number of DNs, most of which can be individually identified across animals. Estimates for the total number of DNs in insects range from 200 to 500 bilateral pairs (*Gronenberg and Strausfeld, 1990*; *Okada et al., 2003*; *Staudacher, 1998*; *Gal and Libersat, 2006*; *Cardona et al., 2009*), with a recent study suggesting *Drosophila* may be at the top of this range (*Hsu and Bhandawat, 2016*). Yet, even this largest estimate is substantially less than the approximately 100,000 cells that descend from the brain to the spinal cord in mice (*Liang et al., 2011*).

Recent combinatorial genetic techniques (*Luan et al., 2006*) make it possible to target individual neurons in the fly nervous system for visualization and manipulation (*Aso et al., 2014*; *Wolff et al., 2015*; *Wu et al., 2016*). Here, we applied these techniques to identify individual DNs in *Drosophila* and create a large collection of selective driver lines that will facilitate their future study. This collection enabled us to systematically map the coarse input and output patterns of the DN population. Our results suggest that DNs in *Drosophila* and other insects are organized into three broad pathways. Two direct pathways link specific regions in the brain to motor centers controlling the wings and legs, respectively. A third, convergent pathway couples a broad array of brain neuropils to a large integrative region between wing and leg neuropil that may control both sets of appendages. This organization thus likely reflects both the function of each pathway's member cells and the evolutionary history of winged insects.

## Results

### Identification of individual DNs

Several thousand neurons run through the cervical connective of flies (*Coggshall et al., 1973*), including both descending neurons (DNs) and ascending neurons (ANs). To estimate the number of DNs, we expressed photoactivatable-GFP (PA-GFP; *Patterson and Lippincott-Schwartz, 2002*) under a pan-neuronal promotor (*nsyb*-LexA, *Jenett et al., 2012*, *Suver et al., 2016*), labeled the neurons by illuminating the neck connective (*Figure 1A,B*), and then counted cell bodies in the brain. For convenience, we operationally define a DN as a cell with a process in the neck connective and a soma in the brain, without a strict requirement forthe direction of information flow. We observed a similar pattern of labeling in four animals and estimated the total number of DNs at ~350 on each side of the brain (~700 total) based on the maximum cell body count among preparations (max = 356 cells, mean ± S.D.=321 ± 23 cells, *N* = 4; *Figure 1C,D*).

Our next goal was to characterize systematically the morphology of each DN and create a library of split-GAL4 lines that targeted individual cell types. We discerned the morphology of individual DNs using a large, publically available database of GAL4-driven expression patterns that covers nearly all neurons in the fly central nervous system (*Pfeiffer et al., 2008*; *Jenett et al., 2012*; *Costa et al., 2016*). We then crossed GAL4-lines of interest (*N* = 586 lines, see Materials and methods) to *teashirt*-GAL80 (*Rubinstein et al., 2010*), an operation that substantially reduced expression of cells with a soma in the VNC, revealing DN axon terminals in the VNC (*Figure 1E*). Based on these results, we selected pairs of driver lines that appeared to target the same DN cell type and made these into 'split half' lines, using the relevant promoters to drive expression of either the activation domain (AD) or the DNA-binding domain (DBD) of the split-GAL4 protein (*Luan et al., 2006*). We then screened over two thousand AD/DBD combinations for expression restricted to single DN cell types. About 10% of the combinations produced an expression pattern sparse enough to include in our collection (*Figure 1F*). In some cases, an AD/DBD split-GAL4 combination drove expression in more than one DN and we used the multi-color flip out technique (*Nern et al., 2015*) to stochastically label individual neurons. We identified a total of 190 bilateral pairs of DNs (out of our original estimate of 350), representing at least 98 cell types (*Figure 2*, *Figure 2—figure supplements 1–13*, *Supplementary file 1*). Estimated cell types are fewer in number than total cells identified because some DN types comprise more than one bilateral pair (see below). We note the morphological similarity of DNs we found to neurons previously reported in Drosophila and other insects in *Supplementary file 2*.

Our original split-GAL4 library included ~200 lines, and we selected a subset of these to create a final library of 133 of the sparsest split-GAL4 lines targeting 54 of the 98 identified DN types with little-to-no background expression (*Supplementary file 3*). As a potential genetic control to be used

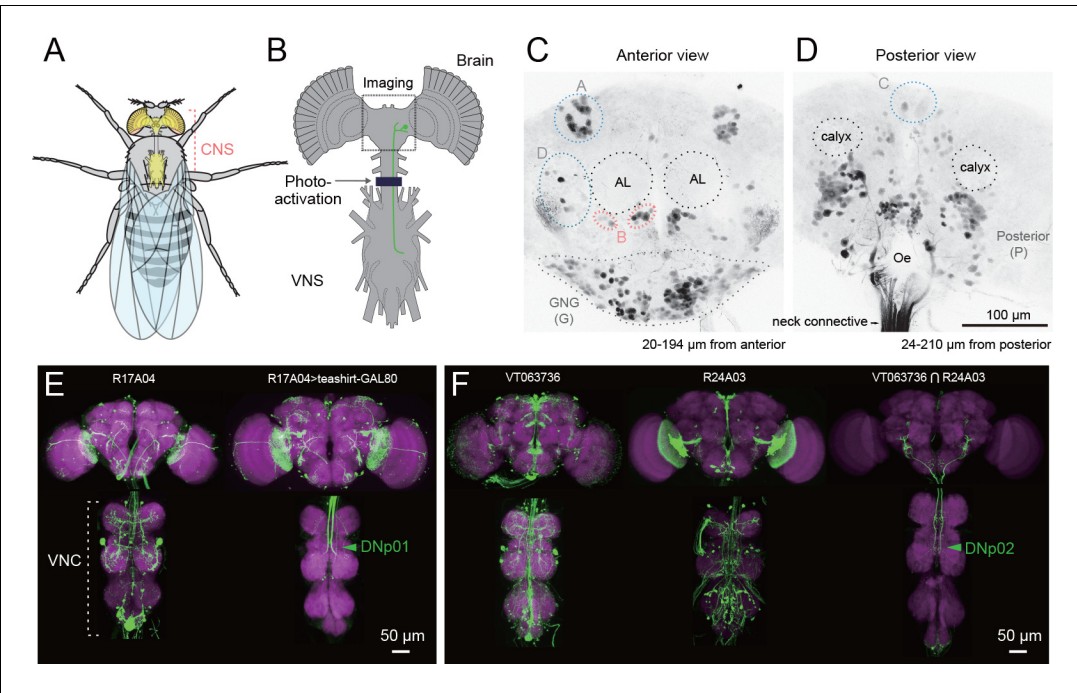

**Figure 1.** Strategy for identifying descending neurons. (**A, B**) The fly central nervous system includes the brain, located in the fly's head, and a ventral nerve cord (VNC), located in the fly's thoracic cavity. These are connected by a population of descending neurons (DNs, example in green), which have cell bodies in the brain. Arrow and dark line indicate area of the neck connective illuminated to selectively label the populations of descending and ascending neurons in a transgenic line pan-neuronally expressing photoactivatable GFP (PA-GFP). Dashed square indicates field of view for imaging results in C, D. (**C, D**) Anterior and posterior views of PA-GFP-labeled DN cell bodies. Black dotted circles represent location of identifiable brain neuropil structures, labeled bilaterally: antennal lobes (AL) and calyx. Blue, light grey, and pink dotted lines enclose separate clusters of DN cell bodies labeled unilaterally: A (anterior dorsal), B (anterior ventral), C (pars intercerebralis), D (outside anterior cluster), G (gnathal ganglion, GNG, shown with dotted line). The uncircled cell bodies in (**D**) are all considered part of the large posterior cluster (P). (**E**) Expression of VNC neurons is suppressed by expression of GAL80 under the *teashirt* promotor. This operation facilitates analysis of DN axonal projection patterns. (**F**) Example of intersection method used to generate split-GAL4 drivers for DNs. VT063736-GAL4 in attP2 (left) and R24A03-GAL4 in attP2 (center) both show expression in DNp02, when crossed to pJFRC2-10XUAS-IVS-mCD8::GFP in attP2 (right). The enhancer fragments from these lines were used to generate the fly line JRC-SS01053 carrying both VT063736-p65ADZp in attP40 and R24A03- ZpGAL4DBD in attP2 (right).

DOI: https://doi.org/10.7554/eLife.34272.002

in future functional studies, we include in this list one line, SS01062, which was made with the same process as the other split-GAL4 lines, but had no expression in the CNS. While making DN split-GAL4 lines, we also serendipitously created lines with sparse expression in VNC inter- and motor neurons, and we list 10 of these in *Supplementary file 4*. These lines (*Supplementary files 3* and *4*) form the basis of our analysis in this paper. Although we did not analyze them in detail, we include a listing of the more broadly expressing DN split-GAL4 lines in *Supplementary file 5*.

DN cell bodies are distributed widely across the brain surface (*Figure 1C–D*). We found 21 DNs (19 types) with somata on the anterior surface, 46 DNs (37 types) on the posterior surface, and 121 DNs (41 types) on the surface of the gnathal ganglia, GNG, a region ventral to the cerebral ganglia, which is separated from them in some insects. (*Figure 1C,D*). Based on somata counts from the PA-GFP experiments, the 190 DNs we describe represent 67% (121/180) of all GNG DNs, 51% (22/41) of anterior DNs, and 35% (46/131) of posterior DNs. The cell body of one neuron is located inside the antenna and is likely the homolog of a previously described campaniform sensory neuron in blowflies (*Nässel et al., 1984*). We developed a simple nomenclature for the DNs in which a prefix identifies it as a descending neuron with a particular cell body position: DNa, anterior dorsal; DNb,

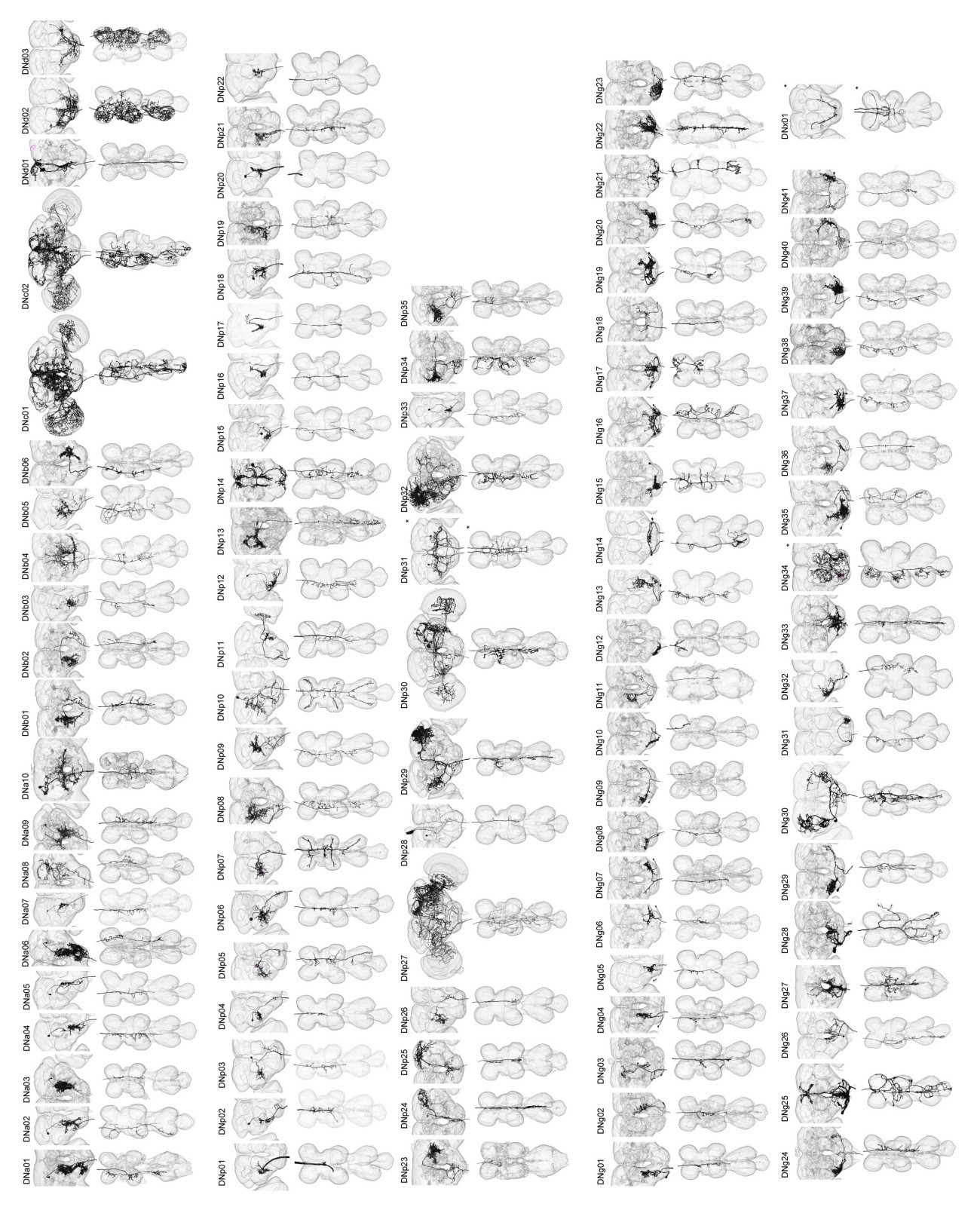

**Figure 2.** Reconstruction of identified descending neurons. Morphology of descending neurons identified in the present study. Neurons (black) and neuropil regions of the brain and VNC are shown (transparent). A total of 98 different cell types are shown. Neurons on both sides of the brain are shown in some cases (asterisk). Segmentation of neuron volume was performed using GAL4 lines with sparse expression and reconstructed with volume rendering. *Figure 2—figure supplements 1–13* shows confocal images masked for individual neurons.

*Figure 2 continued on next page*

*Figure 2 continued*

DOI: https://doi.org/10.7554/eLife.34272.003

The following figure supplements are available for figure 2:

**Figure supplement 1.** Morphology of DNs, group a.
DOI: https://doi.org/10.7554/eLife.34272.004
**Figure supplement 2.** Morphology of DNs, group b.
DOI: https://doi.org/10.7554/eLife.34272.005
**Figure supplement 3.** Morphology of DNs, group c.
DOI: https://doi.org/10.7554/eLife.34272.006
**Figure supplement 4.** Morphology of DNs, group d.
DOI: https://doi.org/10.7554/eLife.34272.007
**Figure supplement 5.** Morphology of DNs, group p (DNp01-11).
DOI: https://doi.org/10.7554/eLife.34272.008
**Figure supplement 6.** Morphology of DNs, group p (DNp12-22).
DOI: https://doi.org/10.7554/eLife.34272.009
**Figure supplement 7.** Morphology of DNs, group p (DNp23-29).
DOI: https://doi.org/10.7554/eLife.34272.010
**Figure supplement 8.** Morphology of DNs, group p (DNp30-35).
DOI: https://doi.org/10.7554/eLife.34272.011
**Figure supplement 9.** Morphology of DNs, group g (DNg01-12)
DOI: https://doi.org/10.7554/eLife.34272.012
**Figure supplement 10.** Morphology of DNs, group g (DNg13-24).
DOI: https://doi.org/10.7554/eLife.34272.013
**Figure supplement 11.** Morphology of DNs, group g (DNg25-34).
DOI: https://doi.org/10.7554/eLife.34272.014
**Figure supplement 12.** Morphology of DNs, group g (DNg35-41).
DOI: https://doi.org/10.7554/eLife.34272.015
**Figure supplement 13.** Morphology of DNs, group x.
DOI: https://doi.org/10.7554/eLife.34272.016

anterior ventral; DNc, pars intercerebralis; DNd, outside cell cluster on the anterior surface; DNg, gnathal ganglion (GNG), DNp, posterior surface of the brain; DNx, outside the brain (see *Figure 2*, *Figure 2—figure supplements 1–13*). Within each prefix designation, cell types are identified by a unique two-digit number. Such a soma-based nomenclature has been adopted elsewhere (*Yu et al., 2010*) and should be more straightforward than one based on the broad locations of either dendrites or terminals, which would be quite byzantine.

Based on morphology, we identified two broad classes of DNs: unique DNs (78 bilateral cell types; *Figure 3A*) that we could tentatively identify as unique bilateral pairs, and population DNs (20 types encompassing 112 cells; *Figure 3B*) that form small groups of cells with nearly identical arbors (*Figure 3—figure supplement 1*). The GAL4 and split-GAL4 lines that targeted the population DNs usually labeled multiple neurons with very similar morphology. We estimated the number of cells within each population type by taking the largest number labeled in a single driver line. In general, population DNs had smaller cell bodies and neurite diameters compared to the unique DNs, and many had soma in the GNG. Populations of DNs with similar morphology and cell bodies in the suboesophageal ganglion were previously observed in the blowfly ('Parallel projecting DNs', *Strausfeld et al., 1984*). However, we cannot exclude the possibility that the DNs we have designated as 'populations' are actually composed of unique bilateral pairs whose morphological similarities mask their individual genetic and functional identity. It is also possible that some of the DNs that we have labeled as 'unique' exist as populations, but we have not yet identified other members of the set. Thus, our classifications should be re-considered in the light of future studies.

## Annotation of individual DN innervation in the brain and VNC

To analyze the coarse connectivity between the brain and VNC, we examined DN projections within neuropils of the central nervous system. To annotate the innervation pattern for each DN (*Figure 4A–D*), we used the recent nomenclature scheme proposed for 41 neuropil regions of the fly

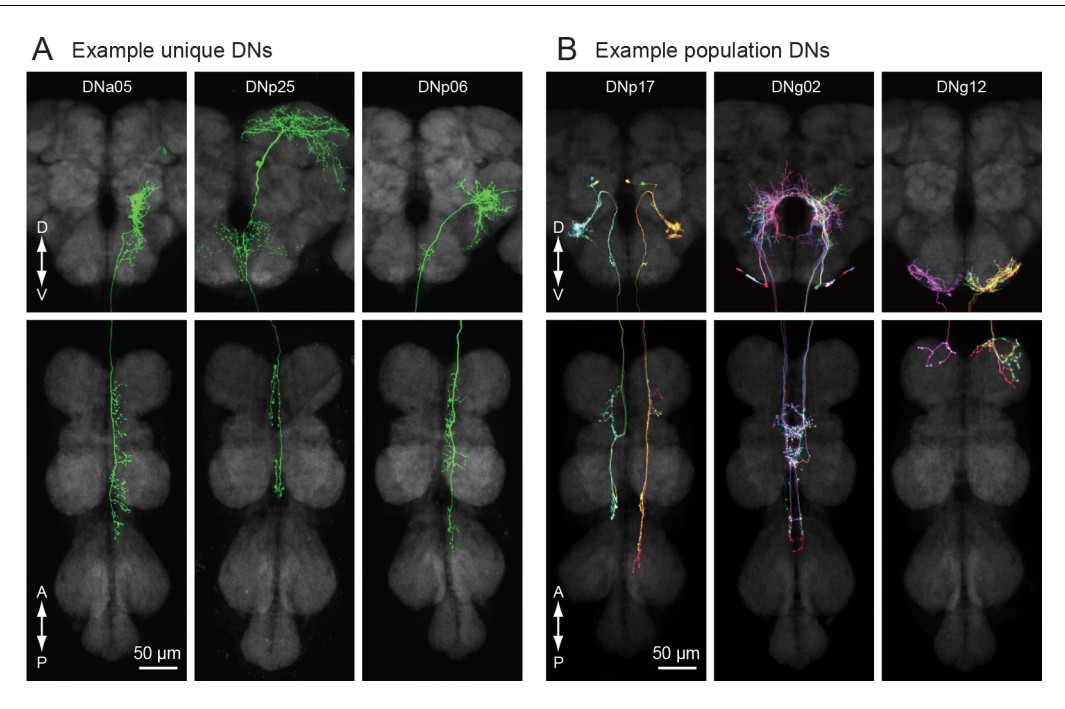

**Figure 3.** Unique and population descending neurons. (**A**) Three example morphologies of DNs that are uniquely identifiable (DNa05, DNp25 and DNp06). Maximum intensity projection images for brain (top) and VNC (bottom) are shown. (**B**) Three examples of population DNs, with individual neurons revealed by multicolor flip-out (DNp17, DNg02 and DNg12). Each neuron of the same DN type shows similar morphology and we do not discriminate individual DNs for these population types.

DOI: https://doi.org/10.7554/eLife.34272.017

The following figure supplement is available for figure 3:

**Figure supplement 1.** Further examples of population DNs.

DOI: https://doi.org/10.7554/eLife.34272.018

brain (*Ito et al., 2014*). The complimentary analysis of DN innervation in the VNC was more challenging, because we lacked a similar standardized atlas for the VNC. *Power (1948)* provided a description of VNC neuropils in *Drosophila melanogaster*; however, these regions have not been defined based on modern molecular markers. Lacking a formal atlas, we defined VNC neuropil region boundaries based on the following criteria (see also *Court et al., 2017*): (1) synaptic areas delineated by nc82 antibody staining, (2) the location of motor neuron dendrites, sensory fiber terminals and tracts, and (3) descriptive information from the literature. These compartmental boundaries were then further refined based on the projection pattern of the DNs as discussed below. In all, we defined 16 VNC neuropils with boundaries as illustrated in *Figure 4E–F*.

Our analysis of the VNC suggests that the tectulum, a neuropil near the dorsal surface associated with flight and other wing-related behaviors (*Power, 1948*; *Murphey et al., 1989*; *Shepherd et al., 2016*), stratifies into a dorsal layer that we propose retain the name 'tectulum' (*Figure 4G*, green) and a more ventral layer we call the 'lower tectulum' (*Figure 4G*, red). This lower region encompasses an area defined by the projections of bristle afferents from macrochaetes on the notum (*Usui-Ishihara and Simpson, 2005*). It is also likely homologous with the bristle neuropil in crickets (*Johnson and Murphey, 1985*) and the anterior ventral association center in locusts (*Tyrer and Gregory, 1982*). As shown below, the segregation of DN terminals in the VNC (e.g. DNp01, p04, p29, p35, g29 and g40) supports the classification of tectulum and lower tectulum as separate functional regions. The arborizations of motor neurons and interneurons within the VNC we identified during the course of DN screening (*Supplementary file 4*) also supports this stratification. For example, the neurites of wing motor neurons and interneurons rarely project below the ventral boundary

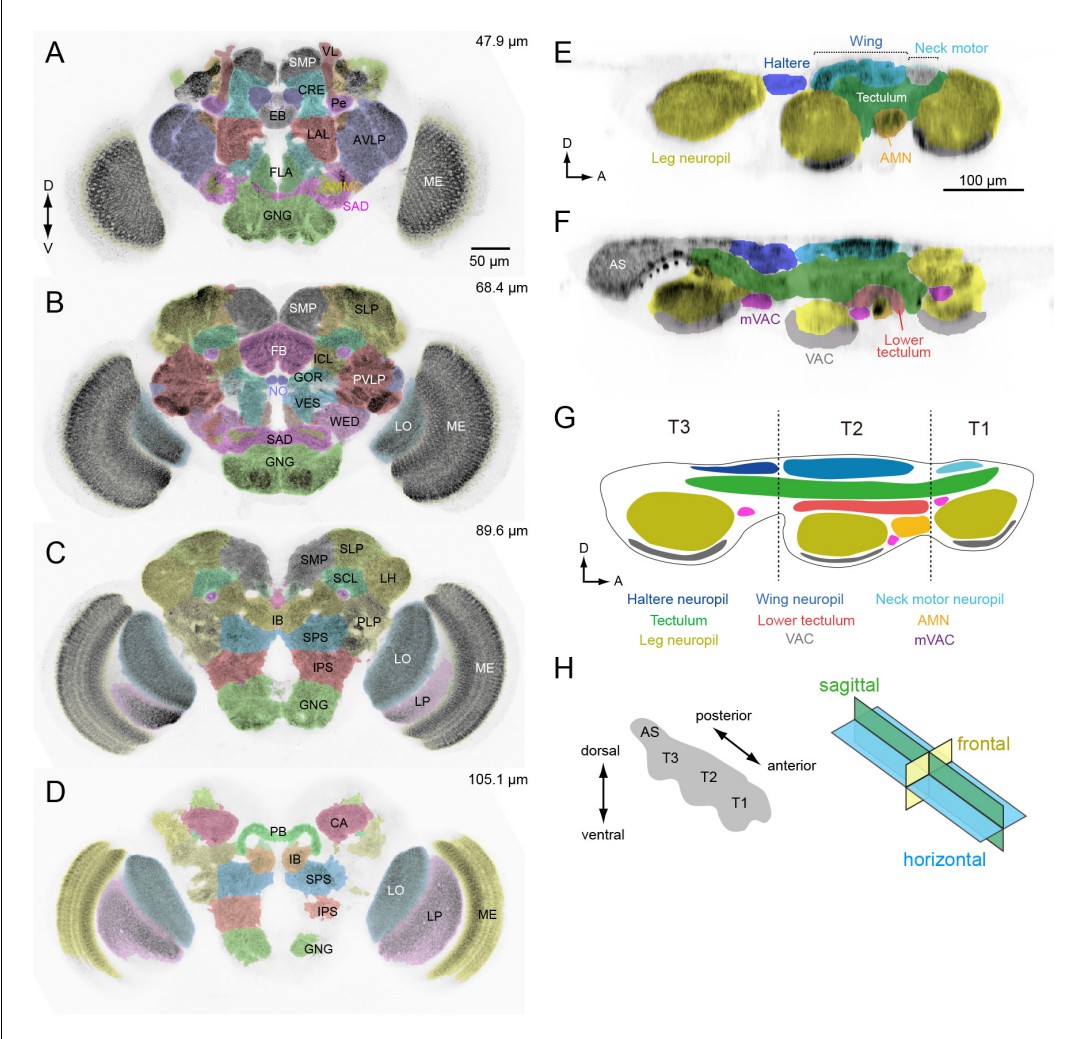

**Figure 4.** Anatomical compartments of the brain and VNC in *Drosophila*. (**A–D**) Identified brain neuropils labeled with different colors superimposed on an aligned confocal image. Depth from the anterior surface is indicated in *top-right* of each image. The data is from Virtual Fly Brain, http://www. virtualflybrain.org/. Neuropil names are from *Ito et al., 2014* and name abbreviations are summarized in *Supplementary file 6*. (**E, F**) Sagittal view of VNC confocal images through a lateral (**E**) and medial (**F**) plane. The colors represent our divisions of the recognized domains in the VNC: AMN (accessory mesothoracic neuropil), AS (abdominal segment), mVAC (medial ventral association center), VAC (ventral association center). (**G**) Schematic of the neuropils in the VNC. T1 (prothoracic segment), T2 (mesothoracic segment), T3 (metathoracic segment). (**H**) The axis and sections used to describe VNC anatomy. The body axis is used.

DOI: https://doi.org/10.7554/eLife.34272.019

The following figure supplements are available for figure 4:

**Figure supplement 1.** Morphology of interneurons innervating the wing neuropil.
DOI: https://doi.org/10.7554/eLife.34272.020
**Figure supplement 2.** Morphology of interneurons innervating the lower tectulum.
DOI: https://doi.org/10.7554/eLife.34272.021

of the tectulum (*Figure 4—figure supplement 1*), whereas we identified new interneurons with arbors that specifically targeted the lower tectulum (*Figure 4—figure supplement 2*).

Our analysis thus far was based on the standard 20x confocal imaging of expression patterns for DN-containing GAL4 and split-GAL4 lines. We re-imaged a subset of these lines with higher magnification (40x, 63x) to resolve finer details. At higher spatial resolution, we observed that DN neurites were either smooth or varicose in appearance, suggesting post-synaptic dendrites and pre-synaptic axonal terminals, respectively (*Peters et al., 1986*; *Römer and Marquart, 1984*). To provide more rigorous molecular evidence for this classification, we examined DN polarity using a reporter

(pJFRC51-3xUAS-Syt::smGFP-HA in su(Hw)attPa) that localizes to presynaptic terminals by labeling of synaptotagmin, a synaptic vesicle-specific protein (*Figure 5A–E*). Localization of the synaptic marker was consistent across preparations (*Figure 5—figure supplement 5*). We analyzed 55 DN types in this fashion and found all varicose terminals were positive for synaptotagmin and all smooth processes were not, at least at the level of the light microscope. This functional interpretation of neurite morphology was also recently validated in *Drosophila* larvae using electron microscopy data (*Cardona et al., 2010*). We thus classified the processes of the remaining 43 DN types as either pre- or post-synaptic, based on morphology alone (*Figure 5F*).

In general, DN arbors in the brain have smooth processes, whereas those in the VNC are varicose (*Figure 5F*), a pattern consistent with a rostral-to-caudal flow of information. However, some DNs possess arbors with varicosities in the brain, indicating that they contact post-synaptic targets before descending to the VNC. It is noteworthy that 78% of the DN types (76/98) have output terminals in the GNG (*Figure 5F*; examples, *Figure 5—figure supplement 1*), whereas 29% have inputs in this region (*Figure 5F*). Although the GNG is fused to the brain in *Drosophila*, comparative and developmental evidence indicates that it is the first member of the chain of segmental ganglia, and thus it is not surprising that so many DNs target circuits in this region.

We observed 43 DN types that have presynaptic terminals (outputs) in the neuropils of the cerebral ganglia (i.e. the brain excluding the GNG) (*Figure 5F*). Six of these (DNp29, p32, g30, p27, c01, and c02) possess extremely large arbors with both inputs and outputs distributed across the cerebral ganglia (*Figure 5—figure supplements 2* and *3*). The remaining 37 DN types have much more specific targets in the cerebral ganglia (*Figure 5—figure supplement 4*). For example, the varicose processes of DNp04 are mostly contained within a single optic glomerulus (*Strausfeld et al., 2007*) (*Figure 5—figure supplement 4A–B*). In 7/37 of the DN types with targeted brain outputs (DNa05, p02, g29, b01, b06, p09), the branches with output synapses appear to be small extensions of those innervating the GNG (*Figure 5—figure supplement 4C*).

*Drosophila* transgenes, including those composing split-GAL4 driver lines, may express stochastically (*Pfeiffer et al., 2010*), *Aso et al., 2014*). We examined stochasticity in our DN split-GAL4 driver lines by inspecting GFP expression in an average of four samples per line (range 2–8). For most of the DN split-GAL4 lines, we did not observe any stochasticity in expression within the number of samples. However, 19 out of 132 lines (14.4%) did show some stochastic expression (one line, SS00735 was excluded from this analysis). *Figure 5—figure supplement 6* shows expression patterns from three samples of three examples lines. Note that for unique DN types, in which only one neuron is targeted per hemisphere, stochastic expression can result in unilateral labeling (SS01558, SS02299, SS02383, SS01061, SS02392, SS02382, SS02394, SS00865; Figure 5—figure supplement 6C), no expression (SS01080, SS02292, SS00730, SS01060, SS02278, SS01566, SS00726), or both (SS01567, SS01543, SS01057). Lines in which we observed stochastic expression are marked in *Supplementary file 3*. We note that expression patterns may also be affected by transgene location and properties of the reporter to which the split-GAL4 driver lines are crossed and so should be verified for any particular experiment.

## DN distribution throughout the CNS

Based on our assignment of inputs and outputs (*Figure 5F*), we compiled the number of DN types with processes in each brain and VNC neuropil (*Figure 6*). DNs extend neurites into nearly every neuropil of the brain, although the connectivity is far from uniform. By far, the region innervated by the largest number of DNs is the GNG. As discussed above, DN arbors in the GNG are largely presynaptic, representing a significant output area for descending information (*Figure 6A*, bottom plot). Restricting the analysis to putative inputs, the largest number of DNs receive input in the inferior posterior slope (IPS) (*Figure 6A*, top plot). Thirteen brain neuropils contained no DN process from the 190 DN cells we identified. Between these two extremes, we found DN neurites in 28 defined neuropil regions, exhibiting a roughly exponential distribution (*Figure 6A*). We did not observe DN dendrites in the central complex or mushroom bodies, suggesting no direct descending output from these high-order, integrative regions.

Most (64%) of the DNs descend within the neck connective ipsilateral to their cell body (121 DNs, 54 types vs. 69 DNs, 44 types, contralateral). Most brain neuropils and tracts contain both ipsi- and contralateral DNs (*Figure 6—figure supplement 1A–C*). The neuropil with the largest asymmetry in the number of ipsilateral and contralateral DNs is the inferior posterior slope (IPS), where a large

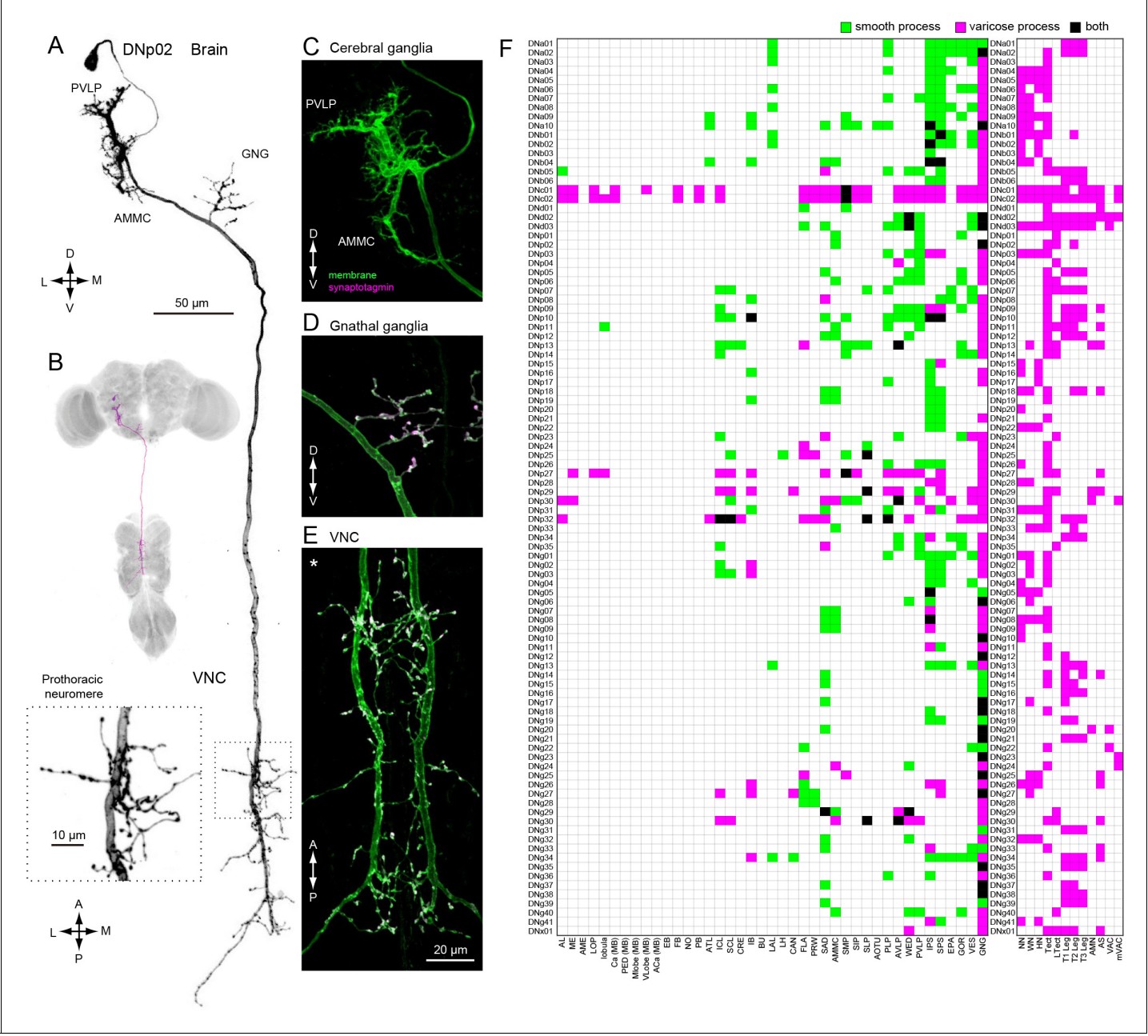

**Figure 5.** Distinguishing DN inputs and outputs. (**A**) The morphology of DNp02. The DN neurites have a smooth appearance in the posterior ventral lateral protocerebrum (PVLP) and anterior mechanosensory motor center (AMMC), and varicose processes in the gnathal ganglia (GNG) and the VNC. Inset shows a magnified view of the DN innervation in the VNC prothoracic ganglia, which have varicose appearance. (**B**) The morphology of DNp02 with nc82 counter-staining. (**C–E**) We determined polarity of DNp02 by cell-specific co-expression of membrane-bound GFP (green) and the presynaptic reporter synaptotagmin-smGFP-HA (magenta). Co-expression (white) is observed in the GNG and VNC, but not in the cerebral ganglion, indicating the DN is post-synaptic in the brain and pre-synaptic in the GNG and VNC. (**F**) Innervation profile of DNs in the brain and VNC. In each row, a filled pixel indicates innervation by the corresponding DN of the CNS neuropil corresponding to the filled column. Green indicates innervation by smooth process, magenta indicates innervation by varicose processes, and black indicates the region receives both types of processes. Smooth and varicose process of DNs are intermingle in the brain. The innervation in the gnathal ganglia (GNG) is mostly by varicose processes. Innervation into the VNC shows varicose endings in all cases.

DOI: https://doi.org/10.7554/eLife.34272.022

The following figure supplements are available for figure 5:

**Figure supplement 1.** DN presynaptic terminals in the gnathal ganglion.

DOI: https://doi.org/10.7554/eLife.34272.023

*Figure 5 continued on next page*

*Figure 5 continued*

**Figure supplement 2.** DN presynaptic terminals in the brain.
DOI: https://doi.org/10.7554/eLife.34272.024

**Figure supplement 3.** DNs in the pars intercerebralis.
DOI: https://doi.org/10.7554/eLife.34272.025

**Figure supplement 4.** DN presynaptic terminals in the brain.
DOI: https://doi.org/10.7554/eLife.34272.026

**Figure supplement 5.** Determining polarity of DNa05 by cell-specific co-expression of a membrane-bound GFP (green) and thepresynaptic reporter synaptotagmin-smGFP-HA (magenta).
DOI: https://doi.org/10.7554/eLife.34272.027

**Figure supplement 6.** Reproducibility of neuronal labeling in Split-Gal4 lines.
DOI: https://doi.org/10.7554/eLife.34272.028

majority of DNs project ipsilaterally. Once in the VNC, the DNs run through eight different tracts, seven of which are readily identifiable with nc82 staining (*Figure 6B*). We counted the largest number of the identified DN axons in the median and intermediate tracts of the dorsal cervical fasciculus (MTD, ITD). Within the VNC, the majority of DNs terminate in the third thoracic segment (61 types, 120 DNs), and only a fraction of DNs extend to the abdominal ganglia (14 types, 16 DNs; *Figure 6C*, *Figure 6—figure supplement 1D*). Overall, the average DN receives input from 2.7 ± 2.0 neuropil regions in the brain and outputs to 1.9 ± 3.6 neuropil regions in the VNC (*Figure 6D–F*).

## Organization of DN outputs

We applied hierarchical clustering to the data in *Figure 5F* to determine the degree to which DN types segregate according to their projection pattern in the VNC (*Figure 7A*). The same data are transformed in *Figure 7B* to visualize the correlations among neuropil regions that emerge from this clustering analysis. With the exception of four DNs that send terminals to nearly every neuropil of the VNC (DNc01, c02, d02 and d03; *Figure 5—figure supplement 3*; *Figure 7—figure supplement 1*), we found a high degree of specificity in which neuropils were targeted by individual DN types (*Figure 7B*). The analysis supports the existence of two strong clusters, one consisting of cells targeting the three segmental pairs of leg neuromeres, and the other targeting the three dorsal neuropils associated with the neck, wing, and halteres, residing in the first, second, and third thoracic segments, respectively. Only a few (<10%) of the DNs that project to the dorsal neuropils (*Figure 7A*, blue), also have terminals in the ventral neuropils (*Figure 7A*, yellow), and vice versa. In fact, DNs with axonal projections to both wing and leg neuropil were quite rare, representing only 6% of the 98 cells types described in our analysis. These include three of the broadly projecting cells described above and three neurons with sparser projections (DNp18, b01, and g17, see *Figure 7—figure supplement 2*). This obvious separation between DNs with axon terminals in the dorsal and ventral VNC neuropil layers suggests independent descending control of wing and leg motor circuits.

Outside of the ~2/3 of DN types that target wing or leg neuropil, most of the remaining third (32/98) selectively targeted one or both of the intermediate neuropil layers, the tectulum and lower tectulum (*Figure 7A*). Six of these DNs targeted the abdominal segment as well (DNp29, p13, d01, g33, p27, and g30). We found only two DNs that terminated exclusively in the smaller association centers: DNg20, which targets the AMN and VAC (*Figure 7—figure supplement 3*), and DNg23, which targets mVAC. These three small neuropils receive input from sensory afferents (*Power, 1948*; *Boerner and Duch, 2010*). We also found one very unique neuron, DNg28, that did not innervate any VNC neuropil but instead sent branching processes along its surface (*Figure 7—figure supplement 4*).

As noted above, many DNs have presynaptic terminals in the brain, most commonly in the GNG. Although to date no cohesive sub-compartmental organization within the GNG has been formalized (*Ito et al., 2014*), the high density of projections there allowed us to test whether this region is structured according to the pattern of DN terminals. We used the nc82 neuropil stain to align to standard templates the confocal images of brain and VNC expression patterns for each of our split-GAL4 driver lines. For the brain, we used the JFRC2013 template, as described previously (*Aso et al., 2014*; *Peng et al., 2011*). For the VNC, we used a VNC template derived from a female CantonS fly

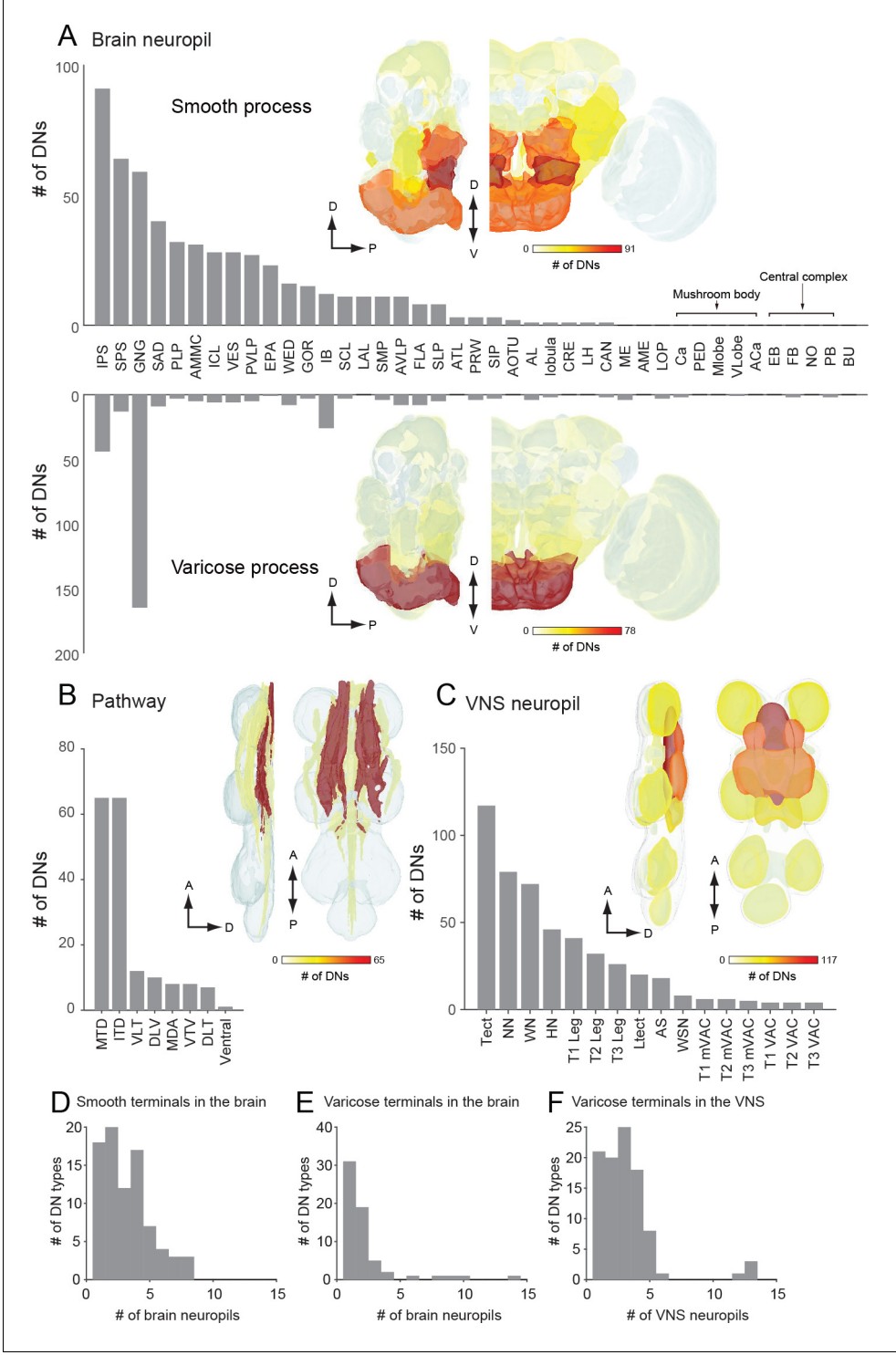

**Figure 6.** The number of identified DNs and their neurite distribution. (**A**) The number of DNs innervating individual brain regions with smooth process (top) and varicose process (bottom). The inset is a heat map of DN innervation in the brain: sagittal and frontal views show brain neuropils in which the number of DNs with processes in each compartment are represented with pseudo-color. Polarity was determined based on their terminal morphology, and confirmed by synaptotamin expression in 55 cell types (see *Figure 5*). Neuropils of the caudal part of the brain, including the superior and inferior posterior slope (SPS, IPS) and GNG, contain smooth processes from the largest number of DNs. The GNG contains varicose processes from the largest number of DNs. The IPS and inferior bridge (IB) also contain varicose processes of many DNs. (**B**) The distribution of DNs

*Figure 6 continued on next page*

*Figure 6 continued*

running through different descending tracts. Inset shows the heat map. The majority of DNs run through either the median tract of the dorsal cervical fasciculus (MTD) or the intermediate tract of the dorsal cervical fasciculus (ITD). The segmented image is modified from *Boerner and Duch, 2010*. Anatomical detail including the position and name for individual tracts are shown in *Figure 3—figure supplement 1*. (**C**) The distribution of DNs innervating individual VNC regions with varicose process. Inset shows the heat map. The number of DNs is greater for the dorsal side than ventral side in the VNC. The tectulum receive the largest descending input. (**D–F**) A histogram of the number of brain (**D–E**) and VNC (**F**) regions innervated by different DN types. Note that panels A-C quantify DN neurons individually, including the number of DNs in a given population type, whereas D-F count DN types, not individual neurons.

DOI: https://doi.org/10.7554/eLife.34272.029
The following figure supplement is available for figure 6:

**Figure supplement 1.** Laterality and extent of DNs axonal projections.
DOI: https://doi.org/10.7554/eLife.34272.030

and performed alignment as described in *Jefferis et al. (2007)*. We then overlaid aligned images of DNs and colored them according to whether they projected to the wing, lower tectulum, or leg layers of the VNC (*Figure 7C–E*). In *Figure 7F*, we overlaid the varicose projections of these same DNs within the GNG, again colored according to their target pattern in the VNC. Our analysis shows that the three classes of DNs target different regions of the GNG and that this pattern recapitulates the dorsal to ventral ordering in the VNC. This analysis suggests that the GNG may be further divided into functional sub-regions, with possible correlation to distinct motor functions that echo those in the VNC.

## Organization of DN inputs

As a complement to our analysis of the DNs based on their outputs in the VNC (*Figure 7*), we also applied hierarchical clustering to the brain neuropils where the cells receive input (see *Figure 5F*). *Figure 8* shows the resulting table of DN types, sorted according to their putative input regions in the brain, along with a resulting matrix of correlation values among the brain neuropils. The analysis suggests the existence of some DN groups and associated clusters of brain neuropils, although the presence of distinct groupings are not as obvious as they were in the analysis of DN outputs in the VNC. One broad cluster consisted of cells with input dendrites in a set of posterior neuropils including the PLP, SPS, IPS, EPA, EPA, VES, and LAL. The two most densely innervated of these regions, SPS and IPS, receive input from visual projection neurons of the lobula plate. PLP also receives input from the lobula plate and contains several optic glomeruli (*Panser et al., 2016*). In contrast, the LAL receives projections from the central complex. Another cluster consisted of cells that innervate the PVLP, WED, GOR, and AVLP. The most densely innervated of these regions, the PVLP, contains the bulk of the optic glomeruli associated with visual projections from the lobula (*Otsuna and Ito, 2006*; *Strausfeld and Okamura, 2007*; *Wu et al., 2016*). The remaining clusters are less obvious, and in general, few coherent patterns emerged from the analysis. For example, it is noteworthy that DNs with dendrites in the GNG tend not to have inputs in other neuropils. For this reason, the GNG does not contribute to an obvious cluster of brain regions. This observation underscores an intrinsic limitation of a cluster analysis. Despite the density of innervation, the importance of a particular brain region will be undervalued if it is associated with DNs that do not receive input from other regions as well. We conclude that DN clustering based on VNC outputs results in groupings that are more distinct from one another (have less overlap) than DN clustering based on their brain inputs.

## Three prominent descending pathways

Our cluster analysis in *Figures 7* and *8* sought an organization of the DNs based on their output and input regions, respectively. However, the relatively ordered pattern of DN projections into motor neuropils associated with specific appendages provides the opportunity to assign putative functions to the brain areas that provide their input. Thus, an alternative strategy - akin to a traditional retrograde backfill - is to visualize the regions of the brain that are innervated by cells targeting specific regions of the VNC (or vice versa, which would represent a orthograde fill). *Figure 9A* shows maps of the number of DNs in the brain targeting each of the eight major neuropils of the VNC.

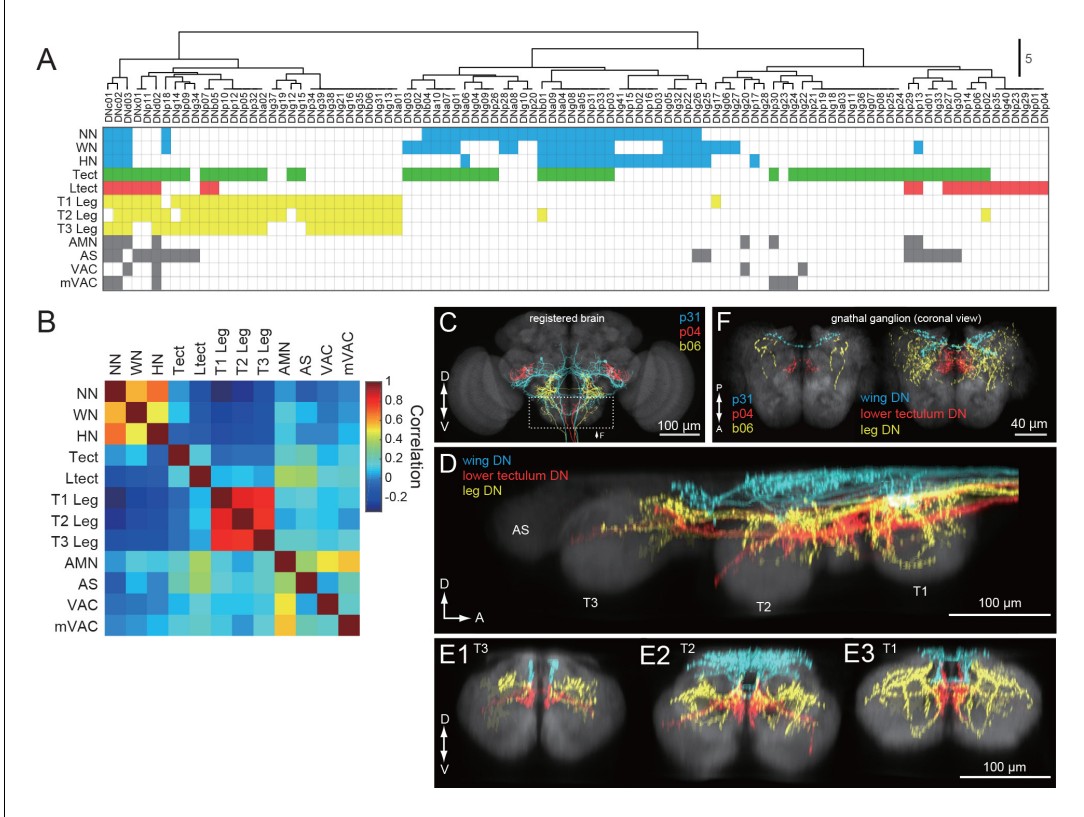

**Figure 7.** DNs differentially address wing/neck/haltere and leg motor systems. (**A**) Clustering of VNC neuropils based on patterns of innervation by each DN. Filled pixels indicate that we observed varicose processes in the neuropil represented by the corresponding row, for the DN in the corresponding column. Pixel color indicates VNC compartment grouping: Dorsal neuropils (blue), tectulum (green), lower tectulum (red), leg neuropils (yellow), or other regions (AMN, AS, VAC, mVAC; gray). (**B**) Autocorrelation matrix of innervation pattern in the VNC. For each pair of VNC compartments, the Pearson's correlation coefficient between DN innervation profiles was calculated. The strongest correlation was amongst compartments within the same grouping (see colors above) but in different segments. (**C–F**) Examination of DN varicose processes in the brain gnathal ganglia. (**C**) Three example DNs from different split-GAL4 lines aligned to a standard brain template and overlaid. Neurons are colored according to which VNC compartments they innervate (wing neuropil, blue; leg neuropils, yellow; lower tectulum, red). (**D**) Sagittal view of axonal projectons within the VNC of a subset of the DN population. (**E**) Transverse view of DN terminals in the different VNC segments: metathoracic (E1), mesothoracic (E2) and prothoracic (E3). (**F**) Horizontal view of DN innervation in the GNG. Magnified view of dashed box in C, shows images of the three example DNs (*left*). A group of 15 DNs for which aligned VNC data were available are also shown in the same view (*right*). The varicose processes of DNs targeting the same compartments in the VNC also form separate clusters in the GNG.

DOI: https://doi.org/10.7554/eLife.34272.031

The following figure supplements are available for figure 7:

**Figure supplement 1.** Putative octopaminergic DNs.
DOI: https://doi.org/10.7554/eLife.34272.032

**Figure supplement 2.** DN projecting to both wing and leg neuropil.
DOI: https://doi.org/10.7554/eLife.34272.033

**Figure supplement 3.** DN projecting to the ventral association center.
DOI: https://doi.org/10.7554/eLife.34272.034

**Figure supplement 4.** DN projecting outside the VNC.
DOI: https://doi.org/10.7554/eLife.34272.035

*Figure 9B* shows the complimentary analysis, in which we show the number of DNs in the VNC that originate from five of the most densely innervated regions in the brain. The complete data set, from which these maps are derived, is presented in *Figure 9C*.

The pattern that emerged in this analysis supports the presence of three major pathways linking the brain to the VNC. Two of the pathways directly connect specific regions in the brain to one each of the two motor neuropils associated with the different forms of adult fly locomotion: flying and

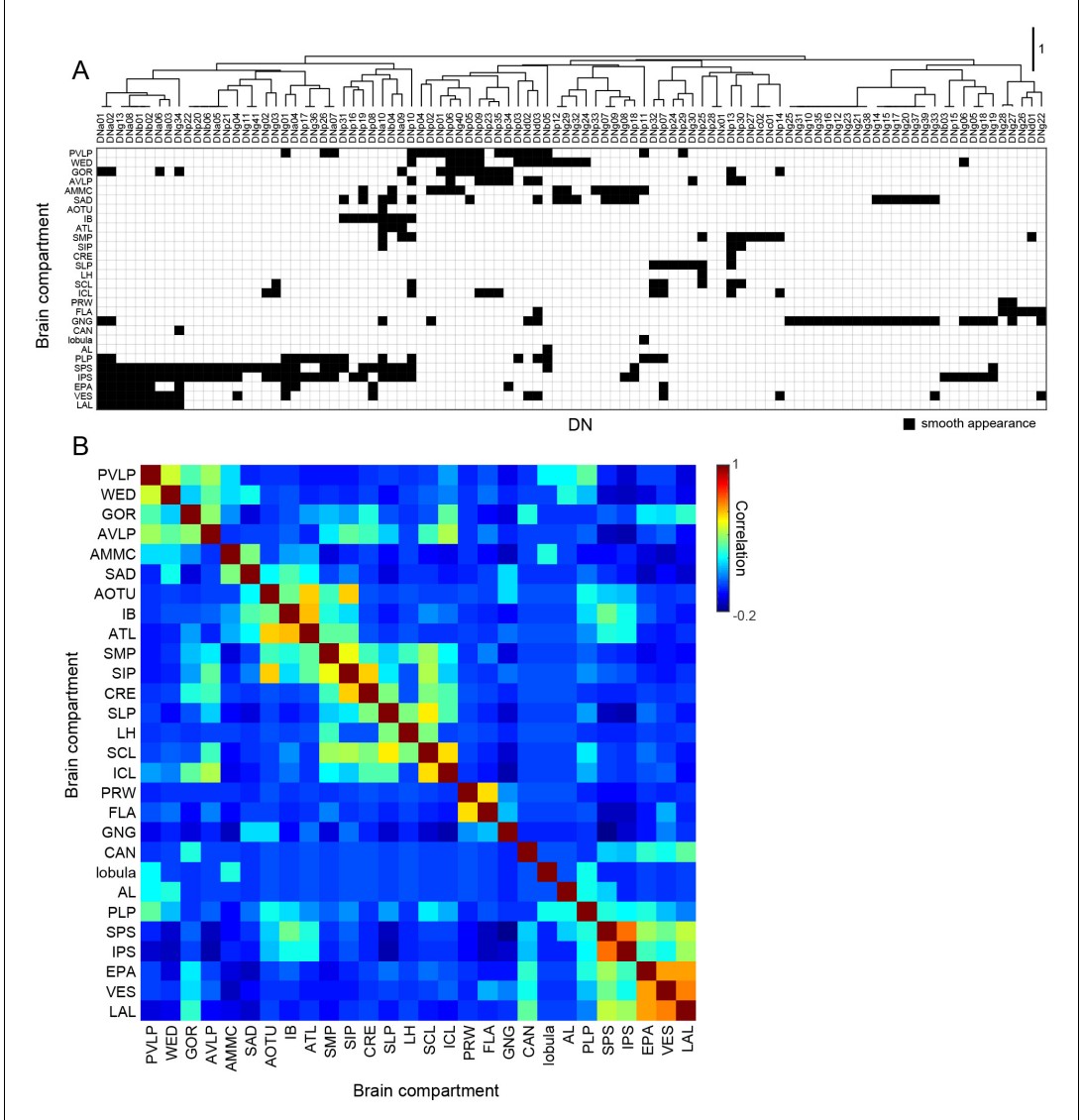

**Figure 8.** Brain innervation by DNs. (**A**) Clustering of brain neuropils based on patterns of DN innervation. Both brain neuropils (rows) and DNs (columns) were sorted by hierarchical clustering based on Pearson's correlation as a metric and average linkage for calculating distances. Only brain compartments with DN innervation are shown. (**B**) Autocorrelation matrix shows the similarity of DN innervation pattern among brain regions. For each pair of brain compartments, the Pearson's correlation coefficient between DN innervation profiles was calculated.
DOI: https://doi.org/10.7554/eLife.34272.036

walking. The first of these is a large number of DNs connecting neuropils of the posterior slope (IPS and SPS) to the dorsal neuropils of the VNC (neck/wing/haltere neuropil and the tectulum). The existence of a major pathway linking the posterior slope and the dorsal VNC neuropils is supported by many prior studies in blowflies (*Strausfeld and Gronenberg, 1990*; *Strausfeld and Lee, 1991*). The second pathway consists of DNs connecting the GNG to the ventral neuropils of the VNC. Note that although many DNs have output synapses in the GNG, the analysis in *Figure 9* is based on inputs. Thus, the enhanced connectivity seen here represents a pathway carrying information from the GNG to the leg neuromeres. Our analysis, which is based on a large number of cells, suggests that these two direct pathways represent dominant conduits of information flow from the brain to the VNC. In contrast to these two direct pathways, the third pathway that emerged from our analysis was a convergent pathway onto the tectulum. In addition to the inputs from IPS and SPS forming the first direct pathway, the tectulum also received input from a large number of neuropils distributed

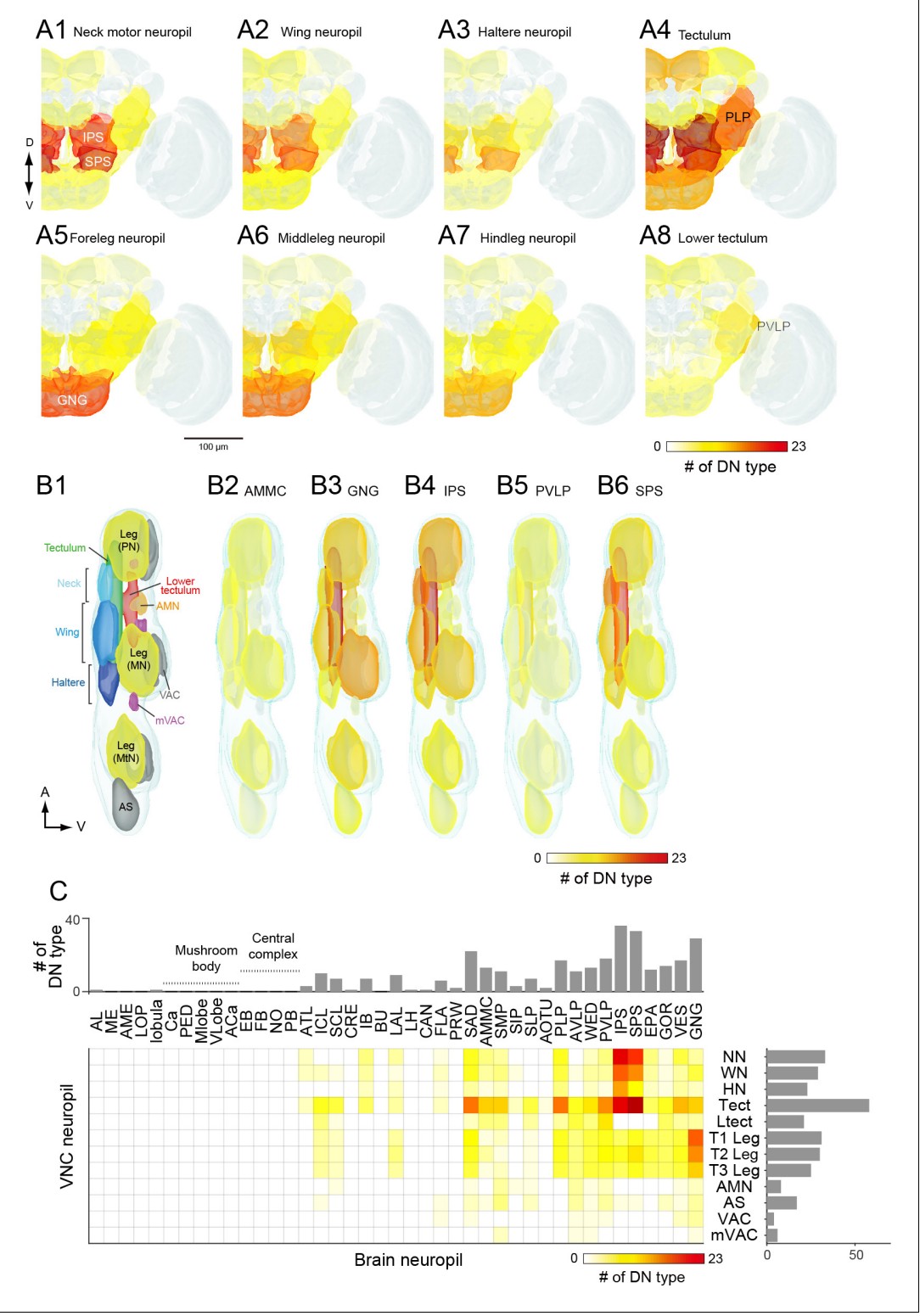

**Figure 9.** DN connectivity between the brain and VNC. (**A**) Dendritic distribution of DNs grouped by output. Heat map colors indicate the number of DN types innervating each brain neuropil for different groups of DNs (A1–A8) defined by their projection to a specified VNC neuropil. The brain innervation pattern is similar among DN groups projecting to the different dorsal VNC neuropils (neck motor, A1; wing, A2; haltere, A3) and among DN groups projecting to the different segmental leg neuropils (foreleg, A5; middle leg, A6; hindleg, A7). The distribution pattern for DNs projecting to the lower tectulum is different from others, with the largest number of DNs

*Figure 9 continued on next page*

*Figure 9 continued*
emanating from the posterior ventral lateral protocerebrum (PVLP). (B) Distribution of DN axonal projections grouped by input. Heat map colors indication the number of DN types innervating each VNC neuropil for different groups of DNs (B2–B6) defined by their projection from a specified a brain neuropil. The VNC atlas is shown in the left panel. Innervation biased for the leg neuropils is observed in DNs from the GNG, for the lower tectulum from the PVLP, and for dorsal neuropils the AMMC, IPS and SPS. (C) The connectivity matrix shows with pseudocolor the number of DNs that innervate both a given brain (columns) and VNC (rows) neuropil. Rich connections are observed from inferior and posterior slope (IPS and SPS) to the dorsal neuropils and from the gnathal ganglia (GNG) to the leg neuropils. For neuropil abbreviations, see *Supplemental file 6*.
DOI: https://doi.org/10.7554/eLife.34272.037

The following figure supplements are available for figure 9:

**Figure supplement 1.** Innervation profile of DNs sorted by innervation clusters in the VNC.
DOI: https://doi.org/10.7554/eLife.34272.038
**Figure supplement 2.** Innervation profile of DNs sorted by innervation clusters in the brain.
DOI: https://doi.org/10.7554/eLife.34272.039
**Figure supplement 3.** Neuronal pathways of DNs.
DOI: https://doi.org/10.7554/eLife.34272.040

throughout the brain, with the notable exception of the mushroom bodies and central complex (*Figure 9C*). Given that the tectulum receives input from so many different brain regions, is sandwiched between the dorsal motor centers and the leg neuromeres, and spans all three thoracic segments, it is tempting to postulate that this neuropil functions in behaviors requiring substantial sensory integration and motor coordination such as courtship or grooming, whereas the direct pathways targeting the dorsal and ventral motor neuropils function more narrowly in locomotion. The pathways projecting to the other VNC neuropils (AS, AMN, VAC, mVAC) are more difficult to characterize because they include fewer of the DNs identified in this study. In general, they appear to follow a convergent pattern in that they receive input from cells originating from an array of different brain neuropils, without any obvious dominant source. One possible exception is the relatively high number of DNs connecting the PVLP with the lower tectulum, which we discuss further below. An analysis visualizing brain innervation patterns based on VNC clusters (*Figure 9—figure supplement 1*), and vice versa (*Figure 9—figure supplement 2*), indicates that the three main pathways identified here are the most prominent organization to emerge from the current analysis.

The axons within the three major pathways distribute among eight descending tracts in a somewhat consistent pattern (*Figure 9—figure supplement 3*). DNs targeting neck, wing, and haltere neuropil descend via one of five tracts: the DLT, MDA, MTD, ITD, and VLT; however, the vast majority of cells within this pathway run within the MTD. The DNs connecting the GNG with the leg neuromeres are distributed much more evenly among a larger number of tracts: DLT, MTD, ITD, VLT, DLV, VTV. Although the DNs projecting to the tectulum arise from a large number of brain nuclei, a very large fraction of them descend via the MTD, with the smaller fraction distributed among all other major tracts.

## Sub-neuropil analyses of DN connectivity

Our study thus far has relied upon a neuropil-to-neuropil analysis of connectivity. Although this is sufficient to reveal large-scale organization of descending pathways, important organization of DN synaptic connectivity likely exists at a finer, sub-neuropil resolution (*Tsubouchi et al., 2017*). Such detailed analyses are possible because we aligned data from each DN type to a standard template (see Materials and methods). A comprehensive analysis of all sub-neuropil DN connectivity is beyond the scope of this paper; however, to demonstrate the importance of sub-neuropil organization and to further explore the sensorimotor pathways identified above, we performed more detailed analyses on three distinct examples. In the sections below, we describe the tract-based motor arrangement of both leg and wing neuropil, as well the organization of DNs descending from the optic glomeruli.

## Tract organization of DN input to leg motor centers

Insect leg neuropil is organized into sub-compartments (*Leise, 1991*); sensory afferents from different classes of mechanoreceptors project to different layers within the leg neuromeres (*Merritt and Murphey, 1992*), and the motorneurons form a myotopic map such that the position of their dendrites along the medial-lateral axis reflects the proximal-distal position of the muscles they innervate in the leg (*Brierley et al., 2012*; *Landgraf et al., 2003*). We observed that the two tracts conveying the largest number of DNs to the leg neuropil, the VLT and MTD, terminate in different strata. The VLT runs along the dorsal surface of the leg neuromeres, near the midline of the VNC, whereas the MTD, together with the DLV, fuse with the oblique tract (OT) running diagonally through each neuromere from its dorsal to ventral surface (*Figure 10A*).

We divided DNs with axon terminals in the leg neuromeres according to the tracts through which they descend (*Figure 10B–F*; see *Figure 6B* and *Figure 9—figure supplement 1* for tract reference). We observed two distinct axon terminal morphologies between these groups. The DNs traversing the VLT and ITD tracts end in bushy axon terminals that are restricted to the dorso-medial zone of each leg neuromere (*Figure 10B,D*). In contrast, the termini of DNs from the MTD or DLV that pass through the OT were more linear, extending across most of the neuromere (*Figure 10E,F*, *Figure 10—figure supplement 1*). In addition, we observed synaptotagmin labeling of DN termini throughout the entire OT (*Figure 10G*, bottom panels). We thus propose that leg motion is controlled by at least two different DN descending systems: one traversing the VLT, which likely contact premotor leg circuits (*n* = 12 DN types) such as those coordinating walking or movement of the proximal leg joints, and one traversing the OT via the MTD and DLV (*n* = 7 DN types), which have the opportunity to control motor neurons innervating muscles spanning all leg joints (*Figure 10—figure supplement 2*).

## Tract organization of DN input to wing motor centers

We identified a total of 29 different DN types supplying the wing neuropil (blue, *Figure 7A*), an area that is defined by the dendritic innervation of wing motor neurons (*Leise, 1991*). Compared to the extensive ramification of wing motor neurons throughout this neuropil (*Vonhoff and Duch, 2010*), the axonal projections of the DNs are notably sparse in most cases (*Figure 11*). Closer inspection of axonal trajectories revealed that wing DNs in the MTD diverge upon entering the VNC. The majority of the DNs that run in the MTD bend ventrally soon after they enter the VNC, whereas a smaller proportion continue along the dorsal surface of the nerve cord before veering ventrally in the second thoracic segment to rejoin the primary MTD group (*Figure 11A–E*). A similar morphology has been observed in the blowfly, *Phormia*, and the dorsal portion we observe resembles the dorso-medial tract in that species (MDT; *Merritt and Murphey, 1992*). In general, the DNs in the ventral MTD route innervate a ventral layer of the wing neuropil, whereas those in the dorsal route terminate above them in a dorsal layer (*Figure 11G*). Thus, our data suggest that the wing neuropil is divisible into two thin dorsal and ventral sublayers, with different DN innervation patterns (*Figure 11H*). This pattern appears to correspond to a stratification of the motor neurons innervating the two functionally distinct types of flight muscle. Motorneurons of the large power muscles, whose stretch-activated oscillations drive the coarse back and forth motion of the wings, have dendrites in the dorsal-most portion of wing neuropil. In contrast, the dendrites of motorneuorns innervating the much smaller steering muscles reside primarily in the ventral layer of wing neuropil (*Figure 11—figure supplements 1* and *2*). We therefore suggest that there may be a separate set of DNs coordinating the power muscles versus the steering muscles.

The sub-neuropil scale analysis of DN projections within the leg and wing neuropils (*Figures 10* and *11*) suggests that DNs may cluster into groups that target different zones within the primary motor neuropils. Further studies will be necessary to determine whether these anatomically identifiable sub-compartments of VNC neuropils are functionally significant.

## Connecting sensory features to motor circuits

To directly examine the correspondence between sensory features and descending pathways, and as a final demonstration of DN sub-neuropil organization, we analyzed DNs that overlap with the optic glomeruli (*Strausfeld and Okamura, 2007*; *Wu et al., 2016*; *Panser et al., 2016*), small, anatomically distinct subcompartments of the ventral and posterior lateral protocerebrum (VLP, PLP)

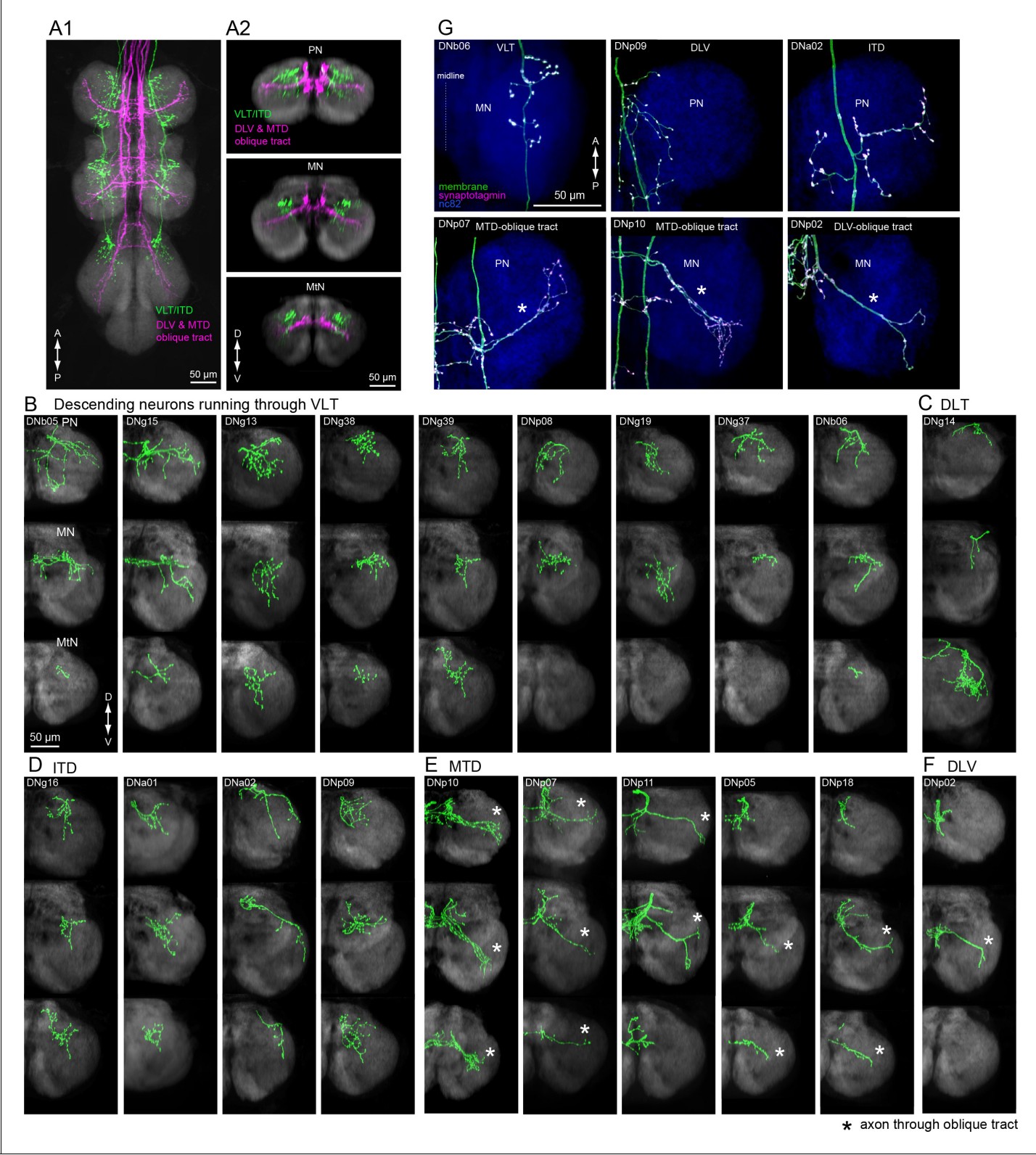

**Figure 10.** Tract-based analysis of DN axonal projection patterns in leg neuropils. (**A**) Major DN types projecting to leg neuropils have different terminal patterns that segregate by descending tract. Horizontal (A1) and frontal (A2) views of overlaid aligned DNs running through VLT, ITD (green), or DLV and MTD (magenta) illustrate these two disparate patterns. (**B–F**) Individual examples of axonal projections to leg neuropil for DNs running through the (**B**) VLT, (**C**) DLT, (**D**) ITD, (**E**) oblique via MTD, or (**F**) oblique via DLV tracts. Transverse sections of the prothoracic (*top*), mesothoracic

*Figure 10 continued on next page*

*Figure 10 continued*

(*middle*) and metathoracic (*bottom*) neuromere are shown. In most cases, the termination zone of axons were similar among the different segments and for DNs withing the same tract. DNs do not innervate the hindleg neuropil in some cases. Note DNg14 was the only DN identified in this study that traverses the DLT tract. (G) Examples of neuronal polarity of DN axonal projection in leg neuropils. The synaptotagmin signal (magenta) was observed in terminals all along the oblique tract (*bottom row*).

DOI: https://doi.org/10.7554/eLife.34272.041

The following figure supplements are available for figure 10:

**Figure supplement 1.** DNs running through the oblique tract.

DOI: https://doi.org/10.7554/eLife.34272.042

**Figure supplement 2.** Potential overlap between DNs and leg motor neurons.

DOI: https://doi.org/10.7554/eLife.34272.043

that each receive inputs from a distinct visual projection neuron type and are suggested to encode different visual features. Individual glomeruli have been shown to act as looming detectors (*Klapoetke et al., 2017*), encode fast (*von Reyn et al., 2014*) or slow (*Wu et al., 2016*) looming velocities, and aid figure-ground discrimination (*Aptekar et al., 2015*). Furthermore, specific activation of the visual neurons projecting to distinct glomeruli elicit different behavioral phenotypes (*Wu et al., 2016*).

We imaged the expression pattern for each of our split-GAL4 lines at a high resolution with an nc82 counter-stain. We identified individual optic glomeruli based on nc82 staining and then scored each of the DNs innervating these areas for strong, weak, or no innervation in each of the 18 identified optic glomeruli (*Figure 12*). Most of the ventral DNs had smooth branches in the optic glomeruli, indicating they receive input there (*Figure 12—figure supplements 1* and *2*). One of the DNs, DNb05 is unusual in that it innervates both optic glomeruli and olfactory glomeruli (*Figure 12—figure supplement 3*). The number of DN outputs from each optic glomeruli were non-uniform, consistent with previous findings in blowflies (*Strausfeld and Okamura, 2007*; *Strausfeld et al., 2007*). Only half of the optic glomeruli are innervated by dendrites of the DNs in this study. Two of the optic glomeruli, LC4 and LC22 (also called LPLC4; *Panser et al., 2016*; *Wu et al., 2016*) are noteworthy in providing divergent output to a large number of DNs (nine and eight cells, respectively, *Figure 12*). LC4 and its neighbors, LPLC1 and LPLC2, form a cluster of densely innervated optic glomeruli in the ventral lateral protocerebrum (*Figure 12—figure supplement 1*), whereas LC22 and a nearby glomerulus, LPLC3, constitute a more posterior group (*Figure 12—figure supplement 2*).

The LC4 glomerulus is of particular interest because it is innervated by the giant fiber (GF, designated DNp01 in our nomenclature). The GF neuron has been analyzed in great detail both anatomically (*Power, 1948*; *Levine and Tracey, 1973*; *Koto et al., 1981*; *Strausfeld and Bassemir, 1983*) and physiologically (*Levine and Tracey, 1973*; *Wyman et al., 1984*; *Mu et al., 2014*; *Fotowat et al., 2009*; *von Reyn et al., 2014*; *von Reyn et al., 2014*) and is critical for a fast mode of escape takeoff in which the fly jumps into the air without prior coordination of its wings (*von Reyn et al., 2014*). Recent work has confirmed that LC4 neurons are functionally presynaptic to the GF and convey information about high-velocity looming expansion, such as that caused by an attacking predator (*von Reyn et al., 2014*). *Milde and Strausfeld, 1990* previously suggested that the GF in a larger species of fly was a member of a cluster of descending neurons with overlapping dendrites that targeted different areas of the VNC. Here, we have identified a group of ventral DNs that all have dendrites in the LC4 glomerulus (*Figure 13*, see also *Figure 12—figure supplement 1*). Although these cells vary widely in their dendritic morphology, all of them arborized within the LC4 glomerulus, and six of them also extend dendrites into the LPLC2 glomerulus, also innervated by GF dendrites.

The ramifications within the LC4 glomerulus differ slightly among these LC4 DNs (*Figure 13B*). For example, DNp04 innervates the entire glomerulus, whereas DNp02 innervation is confined to the ventral portion of the LC4 glomerulus (*Figure 13B*). Localization of dendrites within the optic glomeruli was unexpected, given that previous studies found scant evidence that the glomeruli are organized with retinotopic structure. We found that the dendrites of most DNs innervating optic glomeruli (with the exception of DNp04) are spatially biased within the glomerulus (*Figure 12—figure supplements 1* and *2*), indicating glomeruli may pass on spatial as well as feature information to downstream partners. Three out of the nine LC4 glomerulus DN types project to leg neuropils

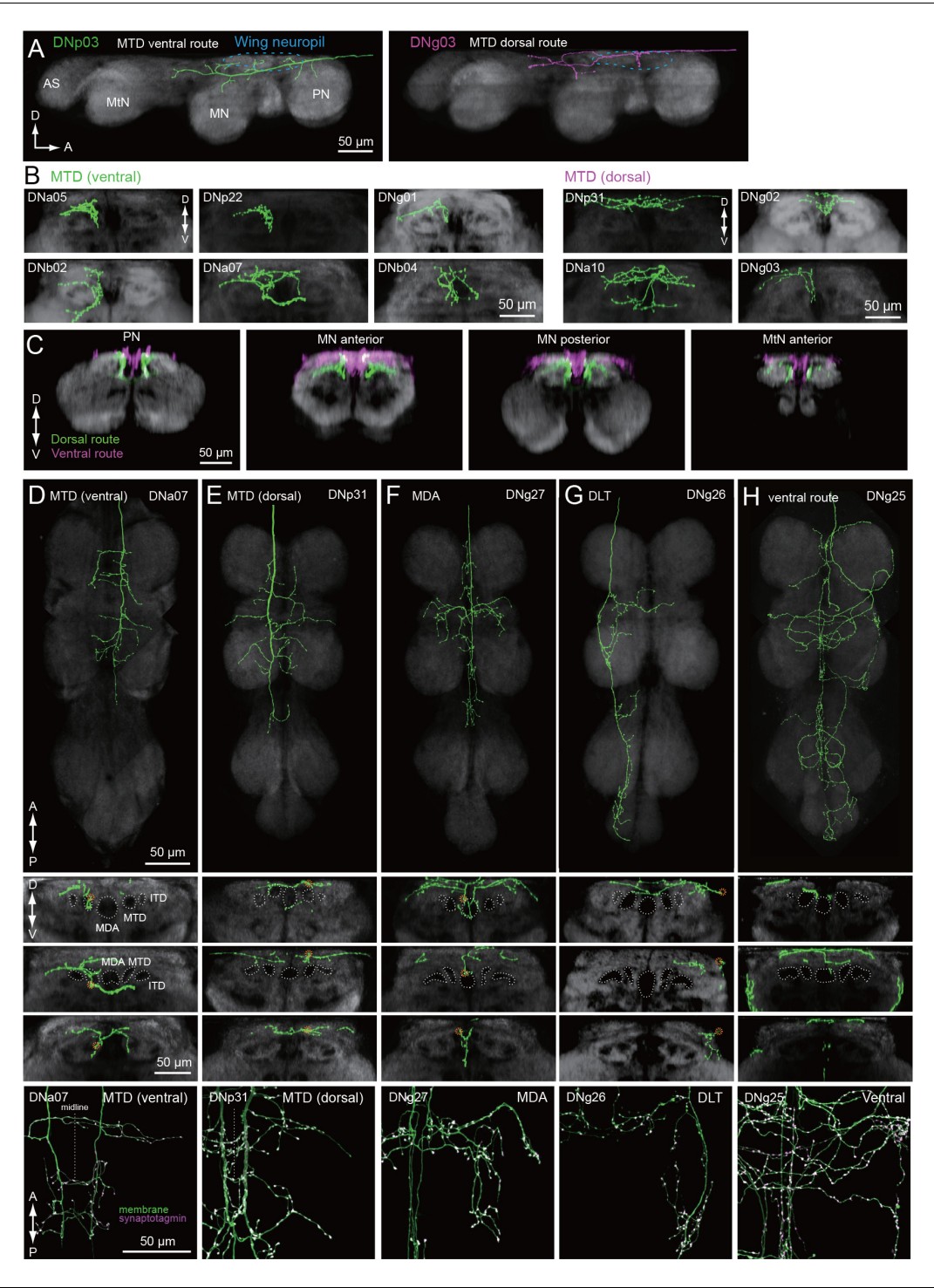

**Figure 11.** Tract-based analysis of DN axonal projection patterns in the wing neuropil. (**A**) Sagittal view of two example DN types targeting wing neuropil (blue dashed line) via the MTD tract. The axon of DNp03 (green) travels ventrally, with the volume of its major axon in the MTD. In contrast, DNg03 runs through the dorsal surface of the VNC from T1 to the middle of T2 segments, and enters the MTD tract in T2 (magenta). (**B–C**) Frontal view of more example axonal projections (**B**) for DNs running through the ventral route (*left*) and dorsal route of the MTD (*right*). Merged images of DNs running through ventral and dorsal MTD route (**C**) shown for a middle section of the prothoracic neuropil, anterior and posterior sections of the mesothoracic neuropil, and an anterior section of the metathoracic neuropil illustrate how the two groups target different sublayers in the wing neuropil. (**D–H**) Individual examples of DN axonal projections in the wing neuropil for DNs running through the (**D**) MTD ventral, (**E**) MTD dorsal, (**F**) MDA, (**G**) DLT, and (**H**) ventral route tracts. Shown are a horizontal view of the whole VNC (*top*), and frontal sections of the prothoracic (*2nd panel*),

*Figure 11 continued on next page*

*Figure 11 continued*

mesothoracic (*3rd*) and metathoracic neuromeres (*4th*). Synaptotagmin labeling is shown at the bottom for the bilateral pair. Note DNg27 is the only DN identified in this study which runs through the MDA.

DOI: https://doi.org/10.7554/eLife.34272.044

The following figure supplements are available for figure 11:

**Figure supplement 1.** Innervation in the wing neuropil of DNs and interneurons.

DOI: https://doi.org/10.7554/eLife.34272.045

**Figure supplement 2.** Innervation in the wing neuropil of DNs and interneurons.

DOI: https://doi.org/10.7554/eLife.34272.046

---

(DNp02, DNp05 and DNp11; *Figure 13C–D*), one projects to the wing neuropil, and the majority (6/9) target the lower tectulum, which is also targeted by the GF. The dendrite of the tergotrochanteral motor neuron and the peripheral synapsing interneuron - both of which are the crucial components for fly escape takeoffs - are present in the lower tectulum and show gap-junction coupling with the GF (*King and Wyman, 1980*; *Blagburn et al., 1999*). In addition, we found several interneurons that connect the lower tectulum with other VNC compartments, including leg and wing neuropil (*Figure 4—figure supplement 2*), suggesting the functional importance of circuits in the lower tectulum, similar to those in the tectulum layer above it, for coordinating actions that involve both leg and wing appendages.

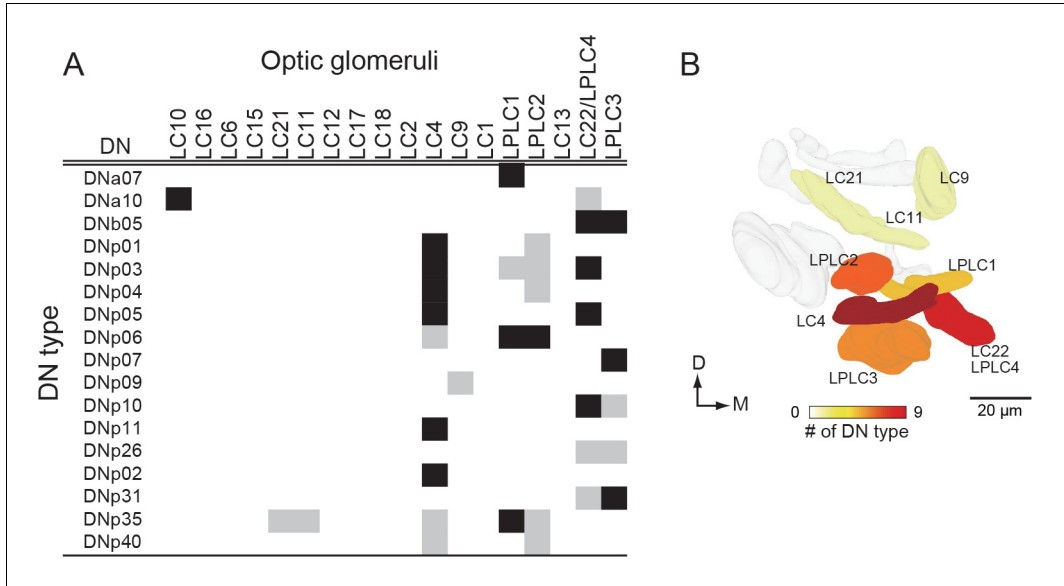

**Figure 12.** DNs from optic glomeruli. (**A**) A matrix indicating DN innervation in the optic glomeruli. Neurite innervation in individual glomeruli was observed with a 63x objective. Black and gray pixel shading represent dense and sparse innervation, respectively. Many DNs were identified that innervate the LC4, and LC22/LPLC4 glomeruli. No DNs were identified that innervated about a half of the glomeruli. (**B**) The number of DNs innervating individual glomeruli is shown as pseudo-color onto the 3D-atlas of optic glomeruli. More DNs were found innervating the more posterior-ventral glomeruli.

DOI: https://doi.org/10.7554/eLife.34272.047

The following figure supplements are available for figure 12:

**Figure supplement 1.** Morphology of DNs from optic glomeruli in the posterior ventral protocerebrum.

DOI: https://doi.org/10.7554/eLife.34272.048

**Figure supplement 2.** Morphology of DNs from optic glomeruli in the posterior lateral protocerebrum.

DOI: https://doi.org/10.7554/eLife.34272.049

**Figure supplement 3.** Morphology of a DN innervating both olfactory and optic glomeruli.

DOI: https://doi.org/10.7554/eLife.34272.050

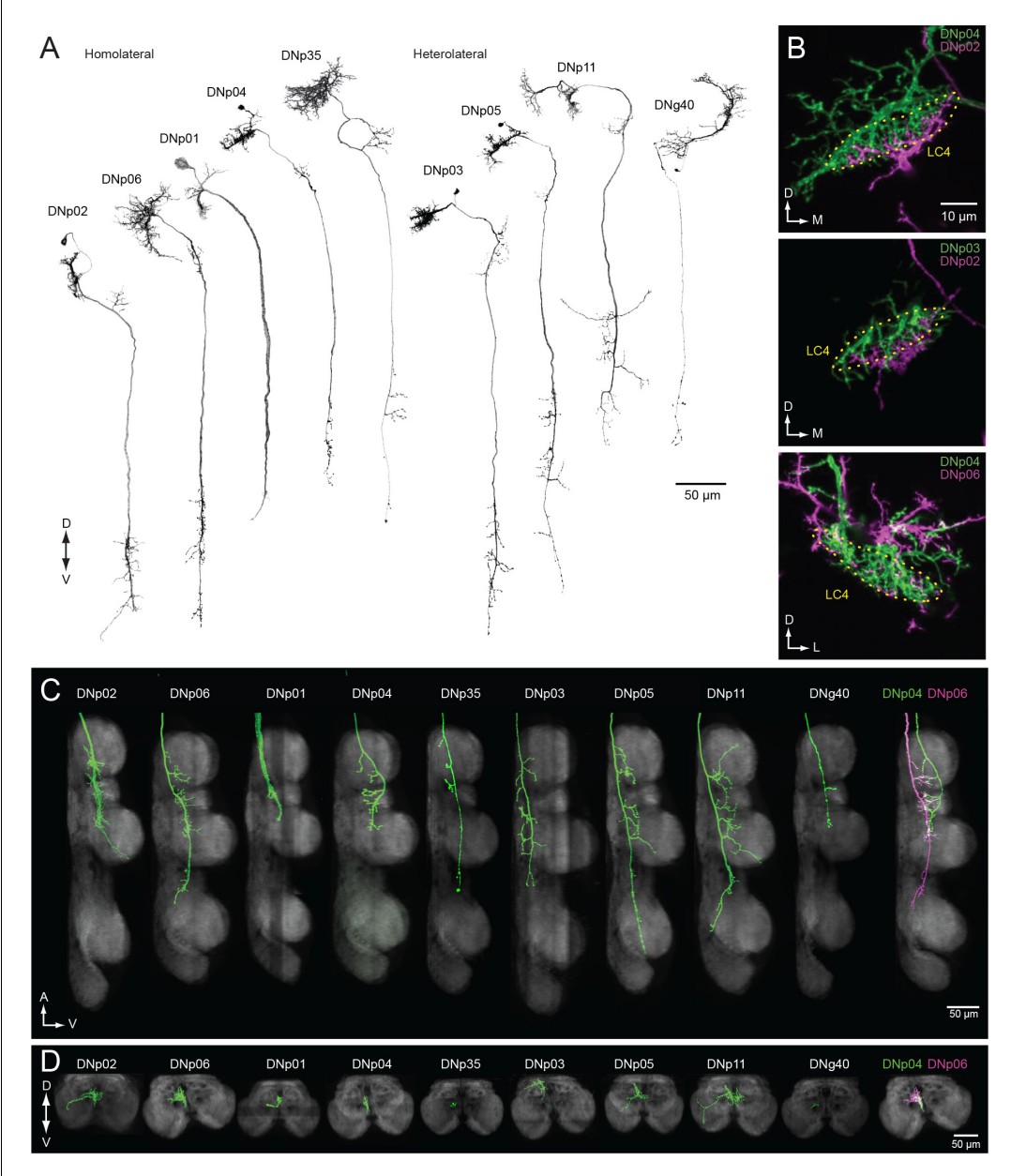

**Figure 13.** DNs forming a dendritic cluster within the LC4 glomerulus project to the lower tectulum. (**A**) The morphologies of DNs which have dendritic innervation to the LC4 glomerulus ('LC4 DNs'). The maximum intensity projection of a confocal stack with 20x objective are shown. All DNs partially share input (LC4 glomeruli) and most of them have axonal projection into the lower tectulum. These DNs are comparable to the 'descending neuron cluster' reported in blowflies (*Milde and Strausfeld, 1990*). (**B**) Simultaneous labeling of two different DNs innervating the LC4 glomerulus, visualized using multicolor flip out (see Materials and methods). Three examples are shown. The shape of the LC4 glomerulus is shown with a dotted line. (**C**) Sagittal view of LC4 DN axonal projections within the VNC. All but one (8/9) have axon terminals in the lower tectulum region of the VNC. One DN does not innervate this region (DNp03). An example of simultaneous labeling of 2 DNs is shown (DNp04 and p06, *right*). (**D**) Frontal view of LC4 DN projections in the mesothoracic neuropil. Projections are focused in the central region of the VNC volume, in the lower tectulum layer.
DOI: https://doi.org/10.7554/eLife.34272.051

## Discussion

In this study, we systematically characterized the organization of DNs, a population of interneurons that conduct information from the brain of a fly to motor centers in the VNC. Our analysis was based on the morphologies of 98 DN cell types, covering 190 bilateral pairs of neurons. To discern DN

morphologies, we segmented individual neurons from driver lines targeting many cells, and we also generated a library of 133 split-GAL4 lines that sparsely target 54 DN types. By registering the morphology of all the DNs with standardized maps of the brain and VNC, we identified three major sensory-motor pathways (*Figures 7* and *9*). One pathway links two neuropils on the posterior slope of the brain (IPS and SPS) to dorsal neuropils associated with the neck, wing, and haltere motor systems, and a second carries neurons with dendrites in the GNG to the leg neuromeres. The third pathway consists of DNs originating from an array of brain neuropils that converge to innervate the tectulum, a long thin region of the VNC sandwiched between the wing and leg motor neuromeres (*Figure 14A,B*).

The simple, tripartite anatomical pattern we observe may reflect both the functional organization of the DNs as well as the evolutionary history of *Drosophila*. With the notable exception of insects (and the mythical horse, Pegasus), all flying animals use a modified foreleg as a wing (*Gatesy and Dial, 1996*; *Thewissen and Babcock, 1991*; *Prokop et al., 2017*; *Seki et al., 2017*). That is, an appendage originally evolved for walking was coopted for flight in pterosaurs, birds, and bats — a fact supported by the fossil record, comparative morphology, and the organization of the underlying motor circuitry (*Gross and Oppenheim, 1985*; *Ryan et al., 1998*). The evolution of flight was quite different in insects, because their wings and associated muscles, did not arise via sacrifice of an entire ancestral leg (*Kukalova-Peck, 1978*; *Dudley, 1994*; *Bruce and Patel, 2018*), and thus the novel aerial mode of locomotion did not strongly compromise the more ancient, terrestrial mode. As a result, insects are unique in possessing two somewhat independent motor systems, a fact that is elegantly manifest in the organization of the VNC and the pattern of DN innervation that we observed: the ventral leg neuromeres of flies resemble those of apterygote hexapods from which they derived, whereas the more recent wing neuropil sits atop the VNC like icing on a cake. We speculate that the GNG-to-leg neuromere descending pathway represents a very ancient pathway and some of its member DNs may have deep homologies with other arthropod taxa, whereas the pathway linking the posterior slope neuropils to the dorsal motor neuropils of the neck, wing, and haltere are more recently evolved within insects.

Many behaviors such as grooming, courtship, take-off, and landing require the simultaneous use of both legs and wings. Thus, insects must have a means of coordinating activity across the two motor systems, a need that arose during or after the evolution of flight. As described more fully below, we speculate that the teculum, and possibly the lower teculum, are neuropils that mediate this functional integration of motor actions between the two systems. The convergence of DNs into the tectulum from such a broad array of brain nuclei may reflect the high degree of sensory integration required to trigger and regulate these more complex, multi-appendage behaviors.

## Estimating DN number

Based on PA-GFP labeling of neurons in the neck connective, we counted ~350 DN pairs. This is within the range of 200–500 DN pairs estimated in other insect species (*Gronenberg and Strausfeld, 1990*; *Okada et al., 2003*; *Staudacher, 1998*; *Gal and Libersat, 2006*), but smaller than a value of ~550 pairs estimated in *Drosophila* based on backfills using a dextran dye (*Hsu and Bhandawat, 2016*). Part of this discrepancy can be explained by the fact that our count excluded several specialized cell populations that were included by *Hsu and Bhandawat (2016)*. These include a set of ~19 pairs of neck motor neurons (*Strausfeld and Seyan, 1985b*), whose axons exit the neck connective posterior to the region we illuminated for PA-GFP photoconversion (*Sandeman and Markl, 1980*), as well as 16 neurons selectively innervating the retrocerebral complex (*Shiga et al., 2000*). We did, however, include one of these cells (DNd01), which innervates both the VNC and retrocerebral complex. Our analysis is also likely an underestimate of the total because the *nsyb*-LexA driver line we used to pan-neuronally express PA-GFP, may not label all neurons. For example, this line does not label the Giant Fiber. It is also possible that certain cells are harder to label using the PA-GFP approach as opposed to dextran backfills. The estimates from the two studies agree quite closely for DNs with cell bodies in the cerebral ganglia (172 in this study vs. 206 in *Hsu and Bhandawat, 2016*). Most of the discrepancy concerns DNs in the GNG group; we counted 180 pairs, only 51% of the number reported by Hsu and Bhandawat. Taking our estimate of 350 as a lower bound and 550 an upper bound, we estimate that the DNs we have described in this study represent between one third and one half of the entire population.

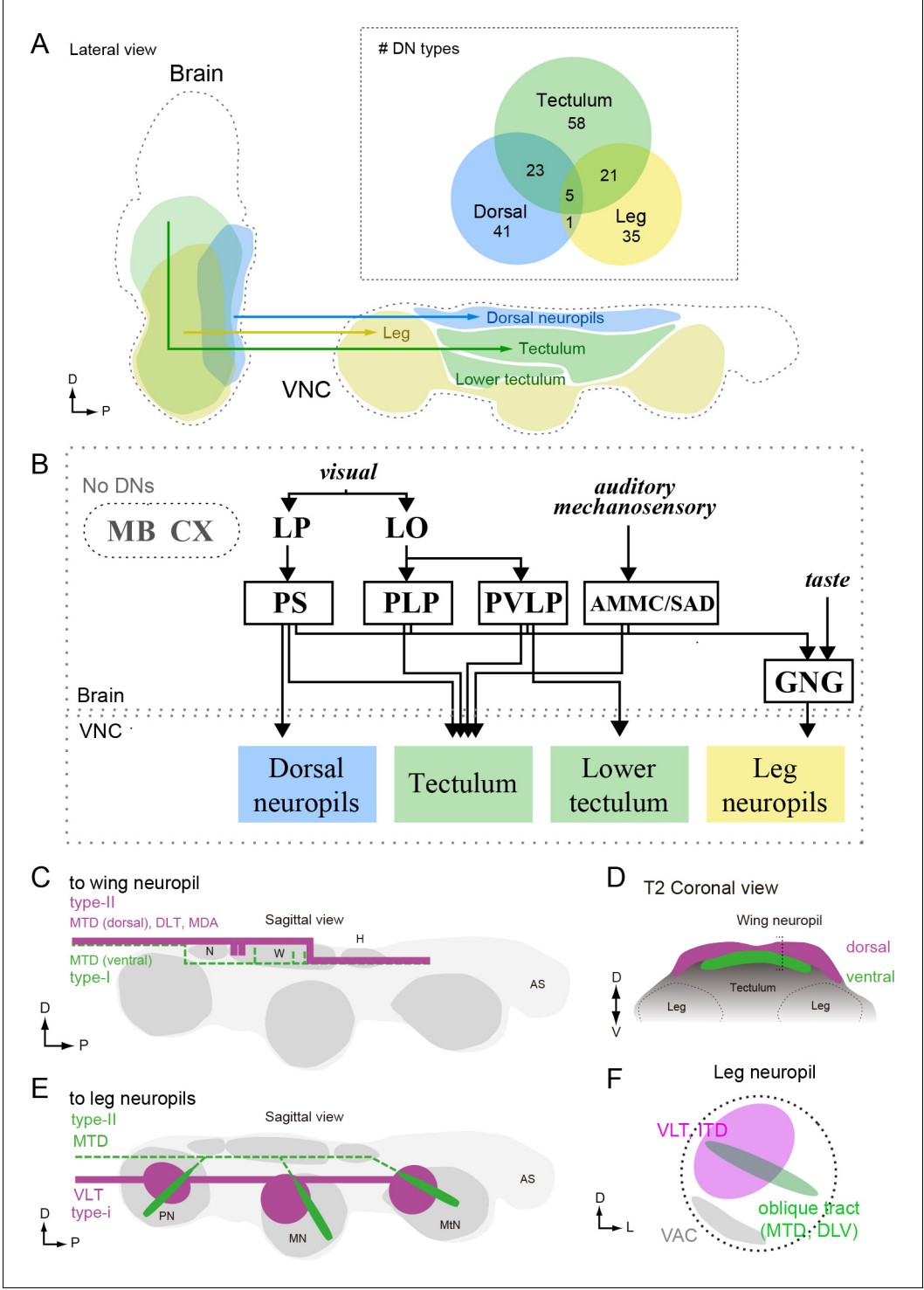

**Figure 14.** Anatomical organization of DNs in *Drosophila*. (**A**) Sagittal view schematic of brain and VNC illustrating the major descending pathways. Gross innervation areas of different DN types are shown with color. Inset shows number of each DN type targeting the three main VNC layers. (**B**) The wiring diagram between the brain and VNC via DNs. Only the major connections are shown. (**C–D**) Schematic of DN axonal projection into wing neuropil shown in sagittal (**C**) and frontal (**D**) views. DN populations that supply axons from the dorsal surface (*type-II*) provide more terminals than those contained within the more ventral MTD tract (*type-I*). See also *Figure 11*. Type-I DNs are more likely to project to the dorsal zone, whereas the type-II DNs are more likely to project to the ventral zone of the wing neuropil. (**E–F**) Schematic showing two types of DN innervation patterns in leg neuropils

*Figure 14 continued on next page*

*Figure 14 continued*

for the whole VNC (**E**) and a single leg neuropil (**F**). The majority of DNs send projections to the medio-dorsal area of leg neuropil (magenta), whereas DNs running through oblique tract via MTD or DLV have fewer terminals and extend to the ventral part of the leg neuropil (green). In most cases, DNs do not innervate the ventral association center (VAC), the ventralmost part of the VNC, which is enriched for afferent sensory projections (**F**).

DOI: https://doi.org/10.7554/eLife.34272.052

Identification of particular DN types in our study relied on the existence of a GAL4-line in the Rubin (*Jenett et al., 2012*) or Vienna (BrainBase, http://brainbase.imp.ac.at/) collection with sparse enough expression to recognize individual DN morphology. Additionally, most of the expression patterns we screened were from female flies, thus our analysis would not include any potential male-specific DNs. As a result, we did not find some DNs that have been reported in other studies, including the Moonwalker Descending Neuron (MDN), which controls backwards walking in flies (*Bidaye et al., 2014*), and pIP10/p2b (*von Philipsborn et al., 2011*; *Kohatsu et al., 2011*), which are involved in the male courtship sequence.

## Pathways to wing neuropil

We found a direct pathway linking the posterior slope of the brain to dorsal VNC neuropils. The posterior slope is innervated by lobula plate tangential cells (LPTCs) projecting from the optic lobe, which are excited by patterns of optic flow resulting from self-rotation (*Krapp and Hengstenberg, 1996*; *Borst et al., 2010*). These optic flow patterns are especially relevant during flight, when the fly is able to move freely about all six degrees of freedom, and it has been suggested that LPTCs mediate both corrective steering maneuvers of the wings (*Wertz et al., 2008*; *Haikala et al., 2013*; *Kim et al., 2015*) as well as gaze stabilization of the head (*Strausfeld et al., 1987*; *Milde et al., 1987*; *Huston and Krapp, 2008*; *Kim et al., 2017*; see also review, *Borst et al., 2010*; *Egelhaaf et al., 2012*). Most of the DNs in this pathway targeted all three segmental dorsal VNC neuropils, which contain neck (T1), wing (T2), or haltere (T3) motor neurons (*Figure 7B*), sensory neuron projections from associated mechanoreceptors (*Chan and Dickinson, 1996*; *Fayyazuddin and Dickinson, 1996*), and premotor interneurons (*Strausfeld and Seyan, 1985b*). DN innervation of all three segmental dorsal neuropils is consistent with recent studies showing that neck and wing movements are highly correlated (*Suver et al., 2016*; *Kim et al., 2017*) and suggests that the DNs of this major posterior slope-to-dorsal neuropil pathway are involved in flight control. This notion is confirmed by recent whole cell recordings from tethered flying flies showing that three members of this population are strongly correlated with compensatory visual responses (*Suver et al., 2016*), and another is involved with spontaneous turns and collision avoidance (*Schnell et al., 2017*).

A similar pathway, in which DNs receiving inputs in the posterior slope target flight neuropil, has been observed in blowflies and flesh flies (*Strausfeld and Lee, 1991*). Strausfeld and Lee contrasted these with other DNs in the protocerebrum that have anterior dendrites near the outputs of the lobula that project to ventral leg neuropils. They suggested that the posterior and anterior DN protocerebral pathways are parallel systems linked to separate photoreceptor channels that process different features of the visual scene (e.g. color vs. motion) and may be loosely analogous to the dorsal and ventral streams of the mammalian visual system (*Strausfeld and Lee, 1991*). Our dataset allowed us to evaluate this hypothesis in *Drosophila* by examining the subset of 42 DNs with dendrites in the protocerebrum (*Figure 15*). In keeping with the observations from large fly species, we did find examples in which a DN with more posterior dendrites (e.g. DNg02, *Figure 15A*, *left*) projected to the dorsal part of the VNC, whereas a DN with anterior dendrites (e.g. DNg13, *Figure 15A*, *right*) projected to the ventral leg neuropils (see also *Figure 15B–C*). We also found that the median location of a DN's dendrites along anterior-posterior axis largely predicted whether its axons targeted dorsal or ventral leg neuropil (*Figure 15D*; although see exceptions DNb01, DNb06, DNp07, and DNp18). However, we found that the dendritic locations of DNs projecting to the dorsal and leg neuropils of the VNC are not segregated into separable, parallel groups, but instead form a continuous pattern of innervation in the protocerebrum. That is, the DN representation is graded in the protocerebrum, at least at the level of resolution of our analysis. Furthermore, the dendritic arbors of many DNs are broad enough that they sample from both anterior and

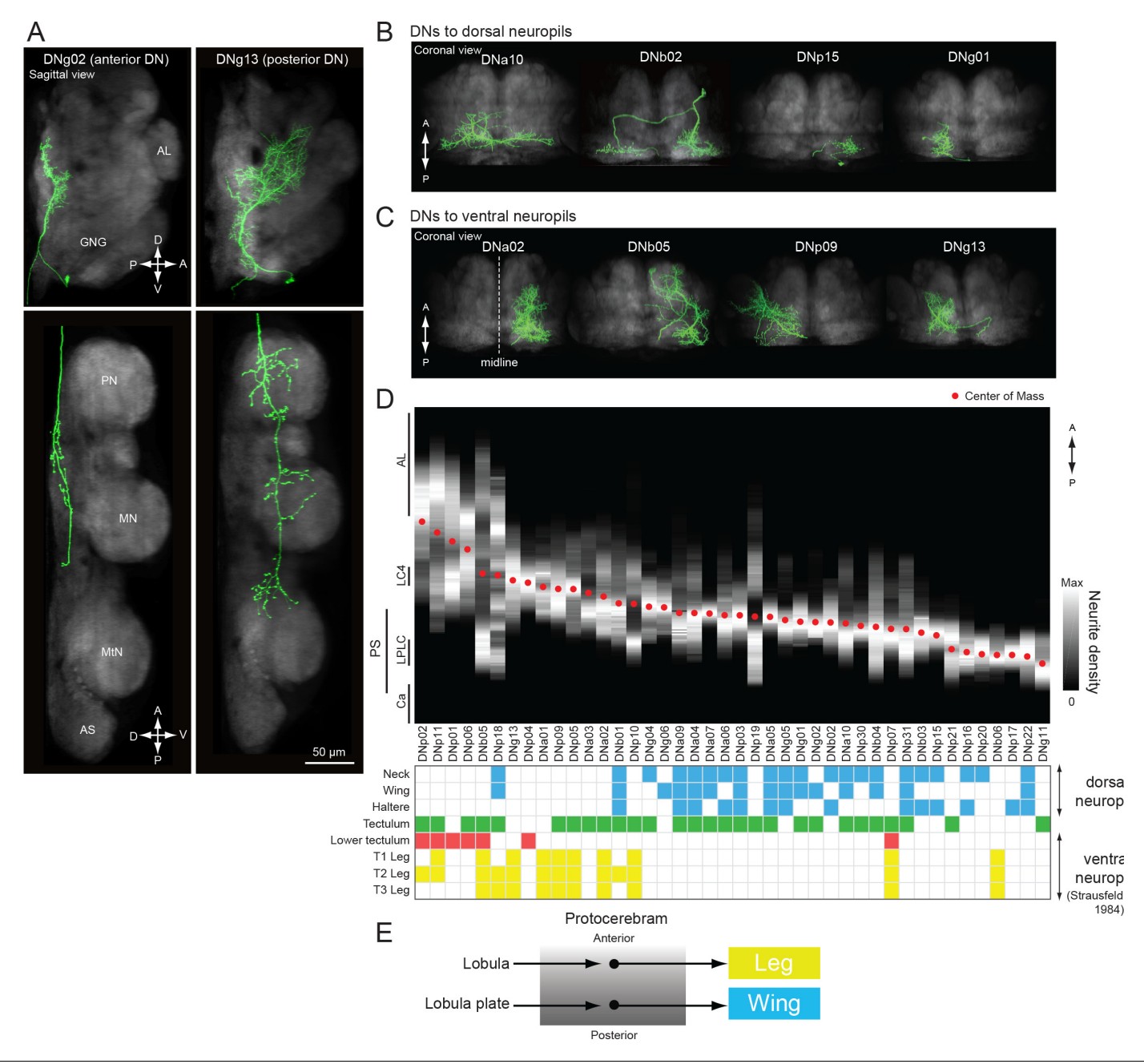

**Figure 15.** Biased representation in the protocerebrum among DNs targeting wing and leg motor areas. (**A**) Morphology of DNs preferentially innervating anterior (DNg02, right) and posterior parts of the brain (DNg13, left). DNg02 projects to dorsal VNC, whereas DNg13 projects to ventral VNC. (**B–C**) Frontal view of brain neurite morphology for four DNs projecting to the (**B**) dorsal or (**C**) ventral neuropils. The dorsal-projecting DNs have neurites limited to the posterior side of the brain, whereas the neurites of ventral-projecting DNs extend to the anterior side of the brain. (**D**) Neurite distribution of the DNs. The relative density of neurites for each DN are shown in gray scale along the anterior-posterior axis based on aligning the data in the registered brain. The red circle indicates the center of mass of the neurite distribution. Along the x-axis, the DNs are arranged in by center of mass position from anterior to posterior. Neurite density was normalized by the maximum value for individual neurons. Table below shows DN projection neuropils in the VNC (blue, dorsal neuropils; yellow, leg neuropils). DNs with innervation toward the ventral side are more likely to project to leg neuropils. (**E**) Schematic of hypothesized information flow in visual descending pathways. The DN dendritic regions in the protocerebrum are not separated, rather there may be a gradient in the preference for axonal projection.

DOI: https://doi.org/10.7554/eLife.34272.053

posterior regions of the protocerebrum, suggesting that many DNs integrate information from both the lobula and lobula plate. Rather than the two separate parallel pathways suggested by *Strausfeld and Lee (1991)* — one carrying visual information from the lobula plate to the wing neuropil and the other carrying information from the lobula to the leg neuropil — we propose that there is a mixing of this visual information in the protocerebrum, possibly in a graded manner along the anterior-posterior axis (*Figure 15E*). A similar divergence and convergence of connectivity has been described in the brainstem of mice. Brainstem nuclei differentially address spinal circuits, forming exclusive connections either with forelimbs, hindlimbs, or both with differing connection strength (*Esposito et al., 2014*).

Among all DNs targeting the wing neuropil, we found evidence for at least two distinct control systems, one entering the neuropil from a dorsal tract and targeting the dorsal and medial portion of the wing neuropil layer, where power muscle motor neuron dendrites reside, and one entering the neuropil from a more ventral tract and invading primarily the ventral and medial wing neuropil, where many steering muscle motor neurons dendrites reside (*Figure 14C,D*). In *Drosophila*, the power muscles comprise two sets of stretch activated muscles attached across the thorax orthogonally (*Dickinson and Tu, 1997*). Alternate deformation of the thoracic cavity by these muscles drives the wing stroke indirectly, powering both flight and courtship song. In contrast, the smaller steering muscles attach to the base of the wing hinge and act directly to coordinate the wing movements that change flight course (*Heide and Götz, 1996*; *Lindsay et al., 2017*), or actuate finer movement, such as the timing of song pulses (*Ewing, 1979*). Our results suggest separate descending control of the power and steering muscle systems. Outside of flight and song, flies perform a wide range of different behaviors with their wings, including grooming (*Szebenyi, 1969*), aggressive displays (*Dow and von Schilcher, 1975*), and preparation for takeoff (*Trimarchi and Schneiderman, 1995*). Although we found that the posterior slope had the largest number of DNs innervating wing neuropil, a wide range of other brain neuropils, including the GNG, VES, PLP, AMMC, SAD, SMP, and LAL, are also connected to the wing neuropil, albeit via a smaller number of DNs. These sparser pathways may be important for coordinating wing motion when the flies are not flying.

## Pathways to leg neuropils

Despite the trend described in the previous section, in which DNs with more anterior dendrites in the protocerebrum tend to target leg neuropil, our analysis (*Figure 9*) found that a different brain region, the GNG, had the strongest DN connectivity to the six ventral neuromeres of the VNC. This was true even after excluding the many DNs whose neurites are presynaptic in the GNG. Indeed, 90% (88/98) of the DN types we found have processes in the GNG, most of which are varicose terminals containing synaptogagmin, and thus likely output terminals. We found that only one-third (29/88) of DNs with processes in the GNG had dendrites in that region, two-thirds of which (18/29) target leg neuropil without any terminals in the dorsal wing, neck, or haltere neuropils.

Given the GNG's evolutionary history as a separate anterior segmental ganglion (*Niven et al., 2008*), it is perhaps not surprising that this neuropil is strongly connected to more posterior motor centers. The suboesophageal ganglion, which includes the GNG (*Ito et al., 2014*), is involved in a variety of behaviors, including walking (*Kien and Williams, 1983*; mantis, *Roeder, 1937*; locust, *Kien, 1990a*,*Kien, 1990b*; stick insect, *Graham, 1979a*,*Graham, 1979b*,*Graham, 1979a* cockroach, *Bässler et al., 1985*), stridulation (*Hedwig, 1986*; *Hedwig, 1994*; *Lins and Elsner, 1995*), flight initiation (*Ramirez, 1988*), head movement (*Altman and Kien, 1979*; *Kien and Altman, 1984*), and respiration (*Ramirez, 1988*; *Otto and Janiszewski, 1989*; *Otto and Hennig, 1993*). However, the GNG has been most specifically implicated in the temporal patterning of walking (*Kien, 1983*; *Kien, 1990a*; *Kien, 1990b*; *Kien and Altman, 1984*; *Kien and Altman, 1992*; *Gal and Libersat, 2006*). For example, both supra- and subesophageal DNs are recruited in the preparatory phase before walking (*Kien, 1990a*), whereas the activity of subesophageal DNs become predominant during the walking phase (*Kien, 1990a*; *Kien, 1990b*).

We found that the terminals of DNs targeting the same layers of the VNC clustered together within the GNG (*Figure 7F*). One intriguing possibility is that these foci represent regions in which efferent copies of descending commands to leg and wing motor centers are available to cephalic sensory circuits. This information could then be integrated directly with other descending commands within the GNG, or reciprocal connections could feed the information back to the cerebral ganglia. The GNG also receives ascending inputs from the leg neuropil (*Tsubouchi et al., 2017*), allowing

further integration within this region of information regarding locomotor state or mechanosensory input. Given that the cerebral ganglia are known to have a strong inhibitory effect on walking in insects, another possibility is that some DN terminals in the GNG are inhibitory. Indeed, a recent study found that 37% of DNs express the inhibitory neurotransmitter GABA, compared to 38% that are cholinergic (*Hsu and Bhandawat, 2016*), and just such an inhibitory pathway from the cerebral ganglia to the GNG has been suggested based on prior behavioral experiments (*Roeder, 1937*; *Gal and Libersat, 2006*). For example, lesion studies have shown that walking persists when the cerebral ganglia are removed and spontaneous bouts are prolonged (*Roeder, 1937*; *Kien, 1983*). In contrast, removal of the GNG reduces spontaneous walking (*Kien, 1983*; *Johnston et al., 1999*), but prolongs flight duration (*Gal and Libersat, 2006*). Thus it is possible that the DN pathway we identified linking the posterior slope to wing neuropil maintains flight and inhibits walking, whereas the pathway linking the GNG to the leg neuropils maintains walking and inhibits flight. Thus, the connections within the GNG may play a critical role in action selection, at least at a coarse level.

We found that DN terminals in the leg neuropils could be sorted into two major types: DNs projecting to the dorso-medial part of each neuromere (type-I) and DNs penetrating through the neuromeres via the oblique tract (type-II) (*Figures 10* and *14E,F*). Their terminal locations suggest that type-I and type-II leg DNs may have different access to leg motor neurons because the dendrites are known to form a rough myotopic map across the leg neuromere, with more proximal leg muscles having more proximal dendrites (*Brierley et al., 2012*). Based on this arrangement, one possible function of the type-I leg DNs is to coordinate the direction of walking, which depends critically on the control of coxal muscles that protract and retract of the entire leg. Indeed, inverse activation of the thoraco-coxal muscle is required for switching from forward to backward walking in stick insects (*Graham and Epstein, 1985*; *Rosenbaum et al., 2010*). In *Drosophila*, moonwalker DNs (MDNs) innervate the dorso-medial part of the leg neuropil and thus are classified as type-I (*Bidaye et al., 2014*). Activation of bilateral MDNs cause backward locomotion, whereas the unilateral activation cause backward turning toward the contralateral side (*Sen et al., 2017*). Type-II DNs running through the oblique tract have the opportunity to contact with the entire array of proximal and distal motor neurons and thus may be important for coordinated action of all leg segments. For example, the jumping part of escape takeoffs may require tension in all leg segments, even though the extrinsic muscle extending the trochanter is the primary actor for the fast takeoff mode (*Trimarchi and Schneiderman, 1993*; *von Reyn et al., 2014*). Consistent with this idea, type-II DNs are abundant in mesothoracic leg neuropil (DNp02, p05, p06 and p11), and it is the middle legs that flies extend during a jump. Similarly, in locust, the descending contralateral movement detector (DCMD, *O'Shea et al., 1974*), which is important for escape behavior, has terminals that resemble type-II and synapses directly on the motor neurons in the neuropil associated with the jumping legs.

## Lower tectulum

We identified a small population of nine DNs specifically projecting to an intermediate zone of the VNC, the lower tectulum, which occupies a volume distinct from wing and leg neuropils and which we suggest can be distinguished from the other intermediate neuropil, the tectulum, that sits above it (*Figure 13*). Neuronal connectivity is not well described in this region, and its function is unknown. However, our observations suggest that, like the tectulum, it is an integrative area involved in both leg and wing control. For example, this region includes dendrites from both the tergotrochanteral leg motor neuron (TTMn) (*Figure 4—figure supplement 2A*) and a branch of a wing motor neuron that we have tentatively identified as III1 (*Figure 4—figure supplement 2E*). The lower teculum also contains the peripheral synapsing interneuron (PSI) (*Bacon and Strausfeld, 1986*), which is presynaptic to motor neurons for the wing depressor muscles (*Figure 4—figure supplement 2B*). The giant fiber (GF) descending neurons that drive a looming-evoked escape takeoff terminate with unbranched axons within the lower tectulum and form gap junctions with the TTMn and PSI (*King and Wyman, 1980*; *Wyman et al., 1984*). We thus surmise that the lower tectulum may play a role during takeoff, which requires coordinated actions of the wings and legs. It is known that there are parallel pathways for take-off behavior in *Drosophila* (*Hammond and O'Shea, 2007*; *Card and Dickinson, 2008b*; *Fotowat et al., 2009*; *von Reyn et al., 2014*), although the anatomical source has not yet been identified. We identified a group of eight unique type DNs, in addition to the GF, whose dendrites overlap with the terminals of visual projection neurons that detect looming (*von Reyn et al., 2014*; *Klapoetke et al., 2017*). Most of these invade the lower tectulum and their

axon terminals share some anatomical features with the GF (*Figure 13*). This population are candidates for parallel pathways for takeoff, as well as other looming-evoked evasive behaviors (*Card and Dickinson, 2008a*, *2008b*; *Card, 2012*), and could represent circuits for wing-leg coordination.

## Candidate descending pathways from higher order centers

We did not find any DNs that originate in the central complex (CX) (*Figures 5F* and *6A*), consistent with studies in other insect species (*Strausfeld, 1976*; *Heinrich, 2002*; *Gronenberg et al., 1995*; *Ito et al., 1998*). Thus, information from the CX must be relayed to motor centers via other brain regions. A prime candidate is the the lateral accessory lobe (LAL), which has dense mutual connections with the CX and, together with the bulb (BU), is considered the CX primary output (*Strausfeld and Hirth, 2013*; *Heinze et al., 2013*; *Lin et al., 2013*; *Wolff et al., 2015*; *Shih et al., 2015*; *Stone et al., 2017*). However, we find many fewer DNs from the LAL than from other regions such as PS, PVLP or AMMC (*Figure 4A*). In other insects such as silk moths, connections between the LAL and the PS are well documented (*Namiki et al., 2014*; *Namiki and Kanzaki, 2016*). In *Drosophila*, connectivity between the LAL and PS is suggested by connectomics studies (*Chiang et al., 2011*; *Ito et al., 2013*; *Yu et al., 2013*; *Shih et al., 2015*) and the morphology of individual neurons connecting these regions has been recently described (*Chiang et al., 2011*; *Costa et al., 2016*, e.g. neuron cluster 31, available from NBLAST web site). Thus, we suggest information processed in the CX may descend to the VNC via a CX-LAL-PS pathway.

We also did not find any DNs originating from the mushroom bodies (MB), important processing areas for olfactory and visual memory (*Heisenberg, 2003*). However, there are 11 DN types innervating the superior medial protocerebrum (SMP), a major target of MB output neurons (*Aso et al., 2014*). The SMP is also well connected with the LAL (*Ito et al., 2013*; *Yu et al., 2013*; *Namiki and Kanzaki, 2016*), which suggests MB output also uses the major descending pathway from the posterior slope via the LAL.

## Behavioral function of DNs

Prior studies in insects have focused on DN function at the single neuron level. Thus, how DNs operate as a population is still unclear. Evidence in insects and other species suggests that motor directives are likely encoded across the DN population rather than in the activity of individual command neurons (*Erickson, 1968*; *Kien, 1983*; *Borgmann and Büschges, 2015*). For example, many DNs are active, albeit with different firing patterns, at the same time during walking in locusts (*Kien, 1990a*, *1990b*), and there are multiple brain locations where electrical stimulation can trigger walking behavior in cockroaches (*Kien, 1983*; *Kien and Williams, 1983*). Also, population vector coding for object direction has been observed in the DNs of dragonflies (*Gonzalez-Bellido et al., 2013*). Zebrafish have also been shown to utilize population coding in the control of locomotion, despite having only ~220 DNs (*Kimmel et al., 1985*; *Metcalfe et al., 1986*) — even fewer than *Drosophila*. In fact, there are very few neurons that fit the rigorous requirements of command neuron (i.e. necessary and sufficient), proposed by *Kupfermann and Weiss (1978)*. Even the giant fibers (a.k.a. DNp01), whose activation drives a stereotyped escape jump in response to looming stimuli, are necessary only for a particular 'fast mode' of takeoff, and the behavioral effect of their activation to naturalistic looming stimuli has been shown to depend on the timing of their spike relative to activity in other descending neurons (*von Reyn et al., 2014*).

Our study found that the VNC areas receiving the largest number of DNs are the dorsal neuropils associated with flight control (neck, wing, haltere neuropils and tectulum). It has been suggested that the number of DNs engaged during a behavior might relate to the precision of the control. In mammals, for example, the number of activated corticospinal tract neurons corresponds to the degree of digital dexterity (*Heffner and Masterton, 1975*, *1983*). It is possible a large DN population target flight neuropils because flight requires a high level of precise control. For example, flies can execute sophisticated rapid aerial turns to evade a looming predator (*Muijres et al., 2014*), movements that are controlled by a combination of adjustments in firing phase of tonically active motor neurons and recruitment of phasically active cells (*Balint and Dickinson, 2001*; *Lindsay et al., 2017*).

In addition to the number of DNs putatively assigned to wing control, our study found that the organization of wing DNs is different than that of the DNs targeting leg neuropil. We identified

several distinct clusters of DNs with nearly identical morphologies and highly overlapped input and output projections, which we refer to as population type DNs because their similar morphology suggests they may function as a group (e.g. DNg01, g02, g03, g05 and g06, *Figure 2—figure supplement 9–12*). In most cases, these population DNs project to the wing neuropil or tectulum and are thus likely involved in flight. In contrast, we found only unique type DNs (identifiable single bilateral pairs) projecting to leg neuropil. This suggests that the strategy for controlling flight and walking may be fundamentally different. Because of the physics involved, even very small changes in wing motion during flight can result in large aerodynamic forces and moments (*Muijres et al., 2014*). The necessity for fine control might account for the greater dependence on population coding in flight as compared to walking. Another difference between flight and walking is the temporal scale required for control. For example, wingbeat frequency is much faster than leg stepping frequency. The control of force generation by wing steering muscles depends on the precise timing of motor neuron spikes (*Tu and Dickinson, 1994*; *Ms and Dickinson, 1996*). The descending input during flight must have the capacity to regulate motor neuron firing phase on a precise temporal scale, a functionality that might be achieved via population coding (*Lehmann and Bartussek, 2017*). Another possibility is that the number of active DNs encodes the magnitude of a command signal to regulate continuous locomotor parameters such as speed. In larval zebrafish and lamprey, for example, more reticulospinal DNs are recruited with increasing swimming frequency (*Brocard and Dubuc, 2003*). Further functional studies will be required to test whether DN encoding of flight and walking commands operates by different principles.

We analyzed the neuronal organization of descending motor pathways in *Drosophila*, with single-cell resolution. The wiring diagram revealed, in a genetically accessible model system, creates a framework for our understanding of how the brain controls behavior. In combination with the *Drosophila* genetic toolkit, the driver lines created in the present study open up the possibility to directly probe the function of individual DNs during natural behavior.

# Materials and methods

## Key resources table

| Reagent type (species) or resource | Designation | Source or reference | Identifiers | Additional information |
|---|---|---|---|---|
| Genetic reagent (*Drosophila melanogaster*) | VT040698-x-VT025718 | this paper | split_gal4.janelia.org:SS02536 | split-GAL4 driver line targeting DNa05 |
| Genetic reagent (*Drosophila melanogaster*) | VT040698-x-VT028606 | this paper | split_gal4.janelia.org:SS01546 | split-GAL4 driver line targeting DNa05, DNa07 |
| Genetic reagent (*Drosophila melanogaster*) | R31H10-x-VT040698 | this paper | split_gal4.janelia.org:SS01560 | split-GAL4 driver line targeting DNa07 |
| Genetic reagent (*Drosophila melanogaster*) | VT023490-x-R80H02 | this paper | split_gal4.janelia.org:SS02393 | split-GAL4 driver line targeting DNa08 |
| Genetic reagent (*Drosophila melanogaster*) | R55D12-x-R72H04 | this paper | split_gal4.janelia.org:SS02370 | split-GAL4 driver line targeting DNb03 |
| Genetic reagent (*Drosophila melanogaster*) | R55D12-x-VT063306 | this paper | split_gal4.janelia.org:SS02552 | split-GAL4 driver line targeting DNb03 |
| Genetic reagent (*Drosophila melanogaster*) | VT019391-x-VT028198 | this paper | split_gal4.janelia.org:SS01051 | split-GAL4 driver line targeting DNb05 |
| Genetic reagent (*Drosophila melanogaster*) | R20C04-x-VT025999 | this paper | split_gal4.janelia.org:SS02631 | split-GAL4 driver line targeting DNb06 |
| Genetic reagent (*Drosophila melanogaster*) | R59A06-x-R64B11 | this paper | split_gal4.janelia.org:SS04161 | split-GAL4 driver line targeting DNc01 |
| Genetic reagent (*Drosophila melanogaster*) | VT012639-x-VT034795 | this paper | split_gal4.janelia.org:SS01570 | split-GAL4 driver line targeting DNd02, DNd03 |
| Genetic reagent (*Drosophila melanogaster*) | VT059225-x-VT057283 | this paper | split_gal4.janelia.org:SS02111 | split-GAL4 driver line targeting DNg10 |
| Genetic reagent (*Drosophila melanogaster*) | VT049363-x-VT057283 | this paper | split_gal4.janelia.org:SS01547 | split-GAL4 driver line targeting DNg10 |

*Continued on next page*

*Continued*

| Reagent type (species) or resource | Designation | Source or reference | Identifiers | Additional information |
|---|---|---|---|---|
| Genetic reagent (*Drosophila melanogaster*) | VT037583-x-R81C11 | this paper | split_gal4.janelia.org:SS01579 | split-GAL4 driver line targeting DNg11 |
| Genetic reagent (*Drosophila melanogaster*) | VT037825-x-VT020379 | this paper | split_gal4.janelia.org:SS02538 | split-GAL4 driver line targeting DNg01 |
| Genetic reagent (*Drosophila melanogaster*) | VT059450-x-VT043626 | this paper | split_gal4.janelia.org:SS02285 | split-GAL4 driver line targeting DNg01 |
| Genetic reagent (*Drosophila melanogaster*) | R38H06-x-VT018689 | this paper | split_gal4.janelia.org:SS01069 | split-GAL4 driver line targeting DNg14 |
| Genetic reagent (*Drosophila melanogaster*) | R53D12-x-VT018689 | this paper | split_gal4.janelia.org:SS04158 | split-GAL4 driver line targeting DNg14 |
| Genetic reagent (*Drosophila melanogaster*) | VT020033-x-VT008483 | this paper | split_gal4.janelia.org:SS02279 | split-GAL4 driver line targeting DNg17 |
| Genetic reagent (*Drosophila melanogaster*) | VT020033-x-VT014208 | this paper | split_gal4.janelia.org:SS00898 | split-GAL4 driver line targeting DNg17 |
| Genetic reagent (*Drosophila melanogaster*) | VT039465-x-VT023750 | this paper | split_gal4.janelia.org:SS02625 | split-GAL4 driver line targeting DNg02 |
| Genetic reagent (*Drosophila melanogaster*) | VT048835-x-R24C07 | this paper | split_gal4.janelia.org:SS01046 | split-GAL4 driver line targeting DNp24 |
| Genetic reagent (*Drosophila melanogaster*) | VT056582-x-R22D06 | this paper | split_gal4.janelia.org:SS01052 | split-GAL4 driver line targeting DNp25 |
| Genetic reagent (*Drosophila melanogaster*) | R21F01-x-R22D06 | this paper | split_gal4.janelia.org:SS01059 | split-GAL4 driver line targeting DNp25 |
| Genetic reagent (*Drosophila melanogaster*) | R27E07-x-VT014958 | this paper | split_gal4.janelia.org:SS01558 | split-GAL4 driver line targeting DNg26 |
| Genetic reagent (*Drosophila melanogaster*) | R13D04-x-R65D05 | this paper | split_gal4.janelia.org:SS01557 | split-GAL4 driver line targeting DNg27 |
| Genetic reagent (*Drosophila melanogaster*) | R59F08-x-R47H03 | this paper | split_gal4.janelia.org:SS04159 | split-GAL4 driver line targeting DNg29 |
| Genetic reagent (*Drosophila melanogaster*) | R42B02-x-VT005005 | this paper | split_gal4.janelia.org:SS02548 | split-GAL4 driver line targeting DNg03 |
| Genetic reagent (*Drosophila melanogaster*) | R61A01-x-R13B05 | this paper | split_gal4.janelia.org:SS02378 | split-GAL4 driver line targeting DNg30 |
| Genetic reagent (*Drosophila melanogaster*) | R14H09-x-R61A01 | this paper | split_gal4.janelia.org:SS02388 | split-GAL4 driver line targeting DNg30 |
| Genetic reagent (*Drosophila melanogaster*) | R14H09-x-R56H09 | this paper | split_gal4.janelia.org:SS01054 | split-GAL4 driver line targeting DNg30 |
| Genetic reagent (*Drosophila melanogaster*) | R61A01-x-VT057280 | this paper | split_gal4.janelia.org:SS01077 | split-GAL4 driver line targeting DNg30 |
| Genetic reagent (*Drosophila melanogaster*) | R14H09-x-VT032228 | this paper | split_gal4.janelia.org:SS02316 | split-GAL4 driver line targeting DNg30 |
| Genetic reagent (*Drosophila melanogaster*) | R42B02-x-VT042734 | this paper | split_gal4.janelia.org:SS02324 | split-GAL4 driver line targeting DNg07 |
| Genetic reagent (*Drosophila melanogaster*) | R42B02-x-R89B01 | this paper | split_gal4.janelia.org:SS01074 | split-GAL4 driver line targeting DNg07 |
| Genetic reagent (*Drosophila melanogaster*) | R42B02-x-VT026005 | this paper | split_gal4.janelia.org:SS01597 | split-GAL4 driver line targeting DNg07 |
| Genetic reagent (*Drosophila melanogaster*) | R42B02-x-VT059427 | this paper | split_gal4.janelia.org:SS02547 | split-GAL4 driver line targeting DNg07 |
| Genetic reagent (*Drosophila melanogaster*) | R73A05-x-VT059427 | this paper | split_gal4.janelia.org:SS02554 | split-GAL4 driver line targeting DNg07 |
| Genetic reagent (*Drosophila melanogaster*) | R42B02-x-VT037574 | this paper | split_gal4.janelia.org:SS02635 | split-GAL4 driver line targeting DNg07, DNg08 |

*Continued*

| Reagent type (species) or resource | Designation | Source or reference | Identifiers | Additional information |
|---|---|---|---|---|
| Genetic reagent (*Drosophila melanogaster*) | R10A12-x-VT004455 | this paper | split_gal4.janelia.org:SS02299 | split-GAL4 driver line targeting DNp01 |
| Genetic reagent (*Drosophila melanogaster*) | R94E01-x-VT031084 | this paper | split_gal4.janelia.org:SS01608 | split-GAL4 driver line targeting DNp10 |
| Genetic reagent (*Drosophila melanogaster*) | R94E01-x-R48E11 | this paper | split_gal4.janelia.org:SS02385 | split-GAL4 driver line targeting DNp10 |
| Genetic reagent (*Drosophila melanogaster*) | VT017683-x-VT031084 | this paper | split_gal4.janelia.org:SS01049 | split-GAL4 driver line targeting DNp10 |
| Genetic reagent (*Drosophila melanogaster*) | VT031084-x-R48E11 | this paper | split_gal4.janelia.org:SS01580 | split-GAL4 driver line targeting DNp10 |
| Genetic reagent (*Drosophila melanogaster*) | R69C11-x-R81D05 | this paper | split_gal4.janelia.org:SS01078 | split-GAL4 driver line targeting DNp20 |
| Genetic reagent (*Drosophila melanogaster*) | R17A04-x-R24A03 | this paper | split_gal4.janelia.org:SS01056 | split-GAL4 driver line targeting DNp28 |
| Genetic reagent (*Drosophila melanogaster*) | R20C03-x-R23C07 | this paper | split_gal4.janelia.org:SS01589 | split-GAL4 driver line targeting DNp28 |
| Genetic reagent (*Drosophila melanogaster*) | R11H10-x-VT040348 | this paper | split_gal4.janelia.org:SS02618 | split-GAL4 driver line targeting DNp29 |
| Genetic reagent (*Drosophila melanogaster*) | R29F12-x-R37G07 | this paper | split_gal4.janelia.org:SS01596 | split-GAL4 driver line targeting DNp03 |
| Genetic reagent (*Drosophila melanogaster*) | R60B12-x-R58E05 | this paper | split_gal4.janelia.org:SS02387 | split-GAL4 driver line targeting DNg30 |
| Genetic reagent (*Drosophila melanogaster*) | R84B12-x-VT048835 | this paper | split_gal4.janelia.org:SS01080 | split-GAL4 driver line targeting DNp04 |
| Genetic reagent (*Drosophila melanogaster*) | VT048835-x-VT017647 | this paper | split_gal4.janelia.org:SS01544 | split-GAL4 driver line targeting DNp04, DNp02 |
| Genetic reagent (*Drosophila melanogaster*) | VT017411-x-VT017647 | this paper | split_gal4.janelia.org:SS02292 | split-GAL4 driver line targeting DNp04, DNp06 |
| Genetic reagent (*Drosophila melanogaster*) | VT019018-x-VT017411 | this paper | split_gal4.janelia.org:SS02256 | split-GAL4 driver line targeting DNp06 |
| Genetic reagent (*Drosophila melanogaster*) | VT029814-x-VT047755 | this paper | split_gal4.janelia.org:SS02276 | split-GAL4 driver line targeting DNp07 |
| Genetic reagent (*Drosophila melanogaster*) | VT023490-x-R38F04 | this paper | split_gal4.janelia.org:SS01540 | split-GAL4 driver line targeting DNp09 |
| Genetic reagent (*Drosophila melanogaster*) | VT047755-x-VT025718 | this paper | split_gal4.janelia.org:SS02310 | split-GAL4 driver line targeting several types of DNa DNs |
| Genetic reagent (*Drosophila melanogaster*) | R22C05-x-R56G08 | this paper | split_gal4.janelia.org:SS00731 | split-GAL4 driver line targeting DNa01 |
| Genetic reagent (*Drosophila melanogaster*) | R75C10-x-R87D07 | this paper | split_gal4.janelia.org:SS00730 | split-GAL4 driver line targeting DNa02 |
| Genetic reagent (*Drosophila melanogaster*) | VT047755-x-R47D05 | this paper | split_gal4.janelia.org:SS02384 | split-GAL4 driver line targeting DNa04, DNa10 |
| Genetic reagent (*Drosophila melanogaster*) | VT061919-x-VT040698 | this paper | split_gal4.janelia.org:SS01552 | split-GAL4 driver line targeting DNa05, DNa07 |
| Genetic reagent (*Drosophila melanogaster*) | VT048835-x-VT017682 | this paper | split_gal4.janelia.org:SS01572 | split-GAL4 driver line targeting DNa05, DNp11 |
| Genetic reagent (*Drosophila melanogaster*) | VT028606-x-R56G08 | this paper | split_gal4.janelia.org:SS01541 | split-GAL4 driver line targeting DNa07 |
| Genetic reagent (*Drosophila melanogaster*) | VT028606-x-R87B09 | this paper | split_gal4.janelia.org:SS01542 | split-GAL4 driver line targeting DNa07 |
| Genetic reagent (*Drosophila melanogaster*) | VT028606-x-VT008675 | this paper | split_gal4.janelia.org:SS01571 | split-GAL4 driver line targeting DNa07 |

*Continued*

| Reagent type (species) or resource | Designation | Source or reference | Identifiers | Additional information |
|---|---|---|---|---|
| Genetic reagent (*Drosophila melanogaster*) | VT013121-x-R45H03 | this paper | split_gal4.janelia.org:SS02383 | split-GAL4 driver line targeting DNb01 |
| Genetic reagent (*Drosophila melanogaster*) | VT049749-x-VT013121 | this paper | split_gal4.janelia.org:SS02542 | split-GAL4 driver line targeting DNb01 |
| Genetic reagent (*Drosophila melanogaster*) | R21F05-x-R93B10 | this paper | split_gal4.janelia.org:SS02396 | split-GAL4 driver line targeting DNb02 |
| Genetic reagent (*Drosophila melanogaster*) | R59B10-x-R21F05 | this paper | split_gal4.janelia.org:SS01600 | split-GAL4 driver line targeting DNb02 |
| Genetic reagent (*Drosophila melanogaster*) | R21F05-x-R21H11 | this paper | split_gal4.janelia.org:SS01060 | split-GAL4 driver line targeting DNb02 |
| Genetic reagent (*Drosophila melanogaster*) | VT027175-x-R83B06 | this paper | split_gal4.janelia.org:SS02395 | split-GAL4 driver line targeting DNc02 |
| Genetic reagent (*Drosophila melanogaster*) | VT001615-x-VT043132 | this paper | split_gal4.janelia.org:SS01545 | split-GAL4 driver line targeting DNd02 |
| Genetic reagent (*Drosophila melanogaster*) | VT001615-x-VT026017 | this paper | split_gal4.janelia.org:SS01575 | split-GAL4 driver line targeting DNd02 |
| Genetic reagent (*Drosophila melanogaster*) | VT012639-x-VT045791 | this paper | split_gal4.janelia.org:SS01569 | split-GAL4 driver line targeting DNd02, DNd03 |
| Genetic reagent (*Drosophila melanogaster*) | VT001615-x-R70C05 | this paper | split_gal4.janelia.org:SS01576 | split-GAL4 driver line targeting DNd02, DNd03 |
| Genetic reagent (*Drosophila melanogaster*) | VT049363-x-VT059225 | this paper | split_gal4.janelia.org:SS02278 | split-GAL4 driver line targeting DNg10 |
| Genetic reagent (*Drosophila melanogaster*) | R81C11-x-R66B05 | this paper | split_gal4.janelia.org:SS02391 | split-GAL4 driver line targeting DNg11 |
| Genetic reagent (*Drosophila melanogaster*) | VT037583-x-VT025598 | this paper | split_gal4.janelia.org:SS01550 | split-GAL4 driver line targeting DNg11 |
| Genetic reagent (*Drosophila melanogaster*) | R66B05-x-R85H06 | this paper | split_gal4.janelia.org:SS01566 | split-GAL4 driver line targeting DNg11 |
| Genetic reagent (*Drosophila melanogaster*) | VT037574-x-VT025598 | this paper | split_gal4.janelia.org:SS02617 | split-GAL4 driver line targeting DNg11 |
| Genetic reagent (*Drosophila melanogaster*) | VT025739-x-VT032280 | this paper | split_gal4.janelia.org:SS02609 | split-GAL4 driver line targeting DNg12 |
| Genetic reagent (*Drosophila melanogaster*) | VT025739-x-VT025999 | this paper | split_gal4.janelia.org:SS02608 | split-GAL4 driver line targeting DNg12 |
| Genetic reagent (*Drosophila melanogaster*) | VT027166-x-VT009857 | this paper | split_gal4.janelia.org:SS02259 | split-GAL4 driver line targeting DNg13 |
| Genetic reagent (*Drosophila melanogaster*) | R88F03-x-VT009857 | this paper | split_gal4.janelia.org:SS01567 | split-GAL4 driver line targeting DNg13 |
| Genetic reagent (*Drosophila melanogaster*) | VT047747-x-VT037825 | this paper | split_gal4.janelia.org:SS02275 | split-GAL4 driver line targeting DNg01 |
| Genetic reagent (*Drosophila melanogaster*) | R60B11-x-R83B04 | this paper | split_gal4.janelia.org:SS01565 | split-GAL4 driver line targeting DNg01 |
| Genetic reagent (*Drosophila melanogaster*) | VT043400-x-VT043662 | this paper | split_gal4.janelia.org:SS02377 | split-GAL4 driver line targeting DNg15 |
| Genetic reagent (*Drosophila melanogaster*) | VT043288-x-VT028153 | this paper | split_gal4.janelia.org:SS01543 | split-GAL4 driver line targeting DNg16 |
| Genetic reagent (*Drosophila melanogaster*) | VT023750-x-VT039465 | this paper | split_gal4.janelia.org:SS02624 | split-GAL4 driver line targeting DNg02 |
| Genetic reagent (*Drosophila melanogaster*) | R42B02-x-VT042835 | this paper | split_gal4.janelia.org:SS02634 | split-GAL4 driver line targeting DNg02 |
| Genetic reagent (*Drosophila melanogaster*) | R42B02-x-R65C10 | this paper | split_gal4.janelia.org:SS01073 | split-GAL4 driver line targeting DNg02 |

*Continued on next page*

*Continued*

| Reagent type (species) or resource | Designation | Source or reference | Identifiers | Additional information |
|---|---|---|---|---|
| Genetic reagent (*Drosophila melanogaster*) | R42B02-x-VT011148 | this paper | split_gal4.janelia.org:SS01561 | split-GAL4 driver line targeting DNg02 |
| Genetic reagent (*Drosophila melanogaster*) | R14F03-x-R24C07 | this paper | split_gal4.janelia.org:SS00732 | split-GAL4 driver line targeting DNp24 |
| Genetic reagent (*Drosophila melanogaster*) | R64B03-x-R29G08 | this paper | split_gal4.janelia.org:SS01602 | split-GAL4 driver line targeting DNg25 |
| Genetic reagent (*Drosophila melanogaster*) | VT057470-x-VT033947 | this paper | split_gal4.janelia.org:SS00923 | split-GAL4 driver line targeting DNp27 |
| Genetic reagent (*Drosophila melanogaster*) | R10A07-x-VT057470 | this paper | split_gal4.janelia.org:SS02257 | split-GAL4 driver line targeting DNp27 |
| Genetic reagent (*Drosophila melanogaster*) | R22H02-x-R20F03 | this paper | split_gal4.janelia.org:SS01061 | split-GAL4 driver line targeting DNg26 |
| Genetic reagent (*Drosophila melanogaster*) | R27E07-x-R14D01 | this paper | split_gal4.janelia.org:SS01064 | split-GAL4 driver line targeting DNg26 |
| Genetic reagent (*Drosophila melanogaster*) | R27E07-x-R20F03 | this paper | split_gal4.janelia.org:SS01593 | split-GAL4 driver line targeting DNg26 |
| Genetic reagent (*Drosophila melanogaster*) | R24F06-x-R45E06 | this paper | split_gal4.janelia.org:SS01063 | split-GAL4 driver line targeting DNg28 |
| Genetic reagent (*Drosophila melanogaster*) | R44A07-x-VT045150 | this paper | split_gal4.janelia.org:SS01564 | split-GAL4 driver line targeting DNg28 |
| Genetic reagent (*Drosophila melanogaster*) | R47H03-x-R59F08 | this paper | split_gal4.janelia.org:SS01075 | split-GAL4 driver line targeting DNg29 |
| Genetic reagent (*Drosophila melanogaster*) | R42B02-x-VT044958 | this paper | split_gal4.janelia.org:SS02633 | split-GAL4 driver line targeting DNg03 |
| Genetic reagent (*Drosophila melanogaster*) | R14H09-x-R13B05 | this paper | split_gal4.janelia.org:SS01588 | split-GAL4 driver line targeting DNp26 |
| genetic reagent (*Drosophila melanogaster*) | VT003476-x-R89A03 | this paper | split_gal4.janelia.org:SS01553 | split-GAL4 driver line targeting DNp26 |
| Genetic reagent (*Drosophila melanogaster*) | R73A05-x-VT012287 | this paper | split_gal4.janelia.org:SS01604 | split-GAL4 driver line targeting DNg07 |
| Genetic reagent (*Drosophila melanogaster*) | R14A01-x-R79H02 | this paper | split_gal4.janelia.org:SS00727 | split-GAL4 driver line targeting DNp01 |
| Genetic reagent (*Drosophila melanogaster*) | R25C08-x-R68A06 | this paper | split_gal4.janelia.org:SS00726 | split-GAL4 driver line targeting DNp01 |
| Genetic reagent (*Drosophila melanogaster*) | VT025392-x-VT057247 | this paper | split_gal4.janelia.org:SS02891 | split-GAL4 driver line targeting DNp11 |
| Genetic reagent (*Drosophila melanogaster*) | R11E07-x-R77F05 | this paper | split_gal4.janelia.org:SS01556 | split-GAL4 driver line targeting DNp15 |
| Genetic reagent (*Drosophila melanogaster*) | R49A07-x-R55A03 | this paper | split_gal4.janelia.org:SS00735 | split-GAL4 driver line targeting DNp16 |
| Genetic reagent (*Drosophila melanogaster*) | R67E08-x-VT025789 | this paper | split_gal4.janelia.org:SS02553 | split-GAL4 driver line targeting DNp17 |
| Genetic reagent (*Drosophila melanogaster*) | VT064490-x-R69C11 | this paper | split_gal4.janelia.org:SS02392 | split-GAL4 driver line targeting DNp18 |
| Genetic reagent (*Drosophila melanogaster*) | VT063736-x-R24A03 | this paper | split_gal4.janelia.org:SS01053 | split-GAL4 driver line targeting DNp02 |
| Genetic reagent (*Drosophila melanogaster*) | VT063736-x-VT017647 | this paper | split_gal4.janelia.org:SS01554 | split-GAL4 driver line targeting DNp02 |
| Genetic reagent (*Drosophila melanogaster*) | VT025392-x-R15E12 | this paper | split_gal4.janelia.org:SS02379 | split-GAL4 driver line targeting DNp02, DNp11 |
| Genetic reagent (*Drosophila melanogaster*) | VT025392-x-VT017647 | this paper | split_gal4.janelia.org:SS02534 | split-GAL4 driver line targeting DNp02, DNp11 |

*Continued*

| Reagent type (species) or resource | Designation | Source or reference | Identifiers | Additional information |
|---|---|---|---|---|
| Genetic reagent (*Drosophila melanogaster*) | R58G11-x-R81D05 | this paper | split_gal4.janelia.org:SS00729 | split-GAL4 driver line targeting DNp20 |
| Genetic reagent (*Drosophila melanogaster*) | R20C05-x-R85H06 | this paper | split_gal4.janelia.org:SS01057 | split-GAL4 driver line targeting DNp20 |
| Genetic reagent (*Drosophila melanogaster*) | R20C03-x-R31B08 | this paper | split_gal4.janelia.org:SS01590 | split-GAL4 driver line targeting DNp28 |
| Genetic reagent (*Drosophila melanogaster*) | R11H10-x-VT033947 | this paper | split_gal4.janelia.org:SS01587 | split-GAL4 driver line targeting DNp29 |
| Genetic reagent (*Drosophila melanogaster*) | R30C12-x-R22D06 | this paper | split_gal4.janelia.org:SS01066 | split-GAL4 driver line targeting DNp03 |
| Genetic reagent (*Drosophila melanogaster*) | R91C05-x-R31B08 | this paper | split_gal4.janelia.org:SS01081 | split-GAL4 driver line targeting DNp03 |
| Genetic reagent (*Drosophila melanogaster*) | R29F12-x-R88C07 | this paper | split_gal4.janelia.org:SS01559 | split-GAL4 driver line targeting DNp03 |
| Genetic reagent (*Drosophila melanogaster*) | R91C05-x-R37G07 | this paper | split_gal4.janelia.org:SS02382 | split-GAL4 driver line targeting DNp03 |
| Genetic reagent (*Drosophila melanogaster*) | R61H01-x-R82C10 | this paper | split_gal4.janelia.org:SS02394 | split-GAL4 driver line targeting DNp32 |
| Genetic reagent (*Drosophila melanogaster*) | VT032898-x-VT048835 | this paper | split_gal4.janelia.org:SS00934 | split-GAL4 driver line targeting DNp04 |
| Genetic reagent (*Drosophila melanogaster*) | R50D07-x-R33H11 | this paper | split_gal4.janelia.org:SS00725 | split-GAL4 driver line targeting DNp04 |
| Genetic reagent (*Drosophila melanogaster*) | VT019060-x-VT003280 | this paper | split_gal4.janelia.org:SS00865 | split-GAL4 driver line targeting DNp05 |
| Genetic reagent (*Drosophila melanogaster*) | VT019018-x-VT017647 | this paper | split_gal4.janelia.org:SS01047 | split-GAL4 driver line targeting DNp06 |
| Genetic reagent (*Drosophila melanogaster*) | VT047755-x-VT003280 | this paper | split_gal4.janelia.org:SS02612 | split-GAL4 driver line targeting DNp07 |
| Genetic reagent (*Drosophila melanogaster*) | VT029814-x-VT003280 | this paper | split_gal4.janelia.org:SS01549 | split-GAL4 driver line targeting DNp07, DNp13 |
| Genetic reagent (*Drosophila melanogaster*) | VT032900-x-VT043145 | this paper | split_gal4.janelia.org:SS01582 | split-GAL4 driver line targeting DNb04 |
| Genetic reagent (*Drosophila melanogaster*) | VT008142-x-VT046808 | this paper | split_gal4.janelia.org:SS01581 | split-GAL4 driver line targeting DNp13, DNp30 |
| Genetic reagent (*Drosophila melanogaster*) | R24A03-x-R74C01 | this paper | split_gal4.janelia.org:SS01062 | split-GAL4 driver line empty brain control |
| Genetic reagent (*Drosophila melanogaster*) | R19G08-x-R47F01 | this paper | split_gal4.janelia.org:SS04528 | split-GAL4 driver line targeting motor neuron b1 (mnb1), a VNC interneuron, with weak expression in DNg02 and possibly DNg06 |
| Genetic reagent (*Drosophila melanogaster*) | R10A12-x-R81E05 | this paper | split_gal4.janelia.org:SS00737 | split-GAL4 driver line targeting DNp20, DNp22, motor neuron b2 (mnb2), and dorsoventral muscle motor neurons (DVM) |
| Genetic reagent (*Drosophila melanogaster*) | R18A05-x-VT045791 | this paper | split_gal4.janelia.org:SS31976 | split-GAL4 driver line targeting an interneuron |
| Genetic reagent (*Drosophila melanogaster*) | VT016973-x-VT002042 | this paper | split_gal4.janelia.org:SS02623 | split-GAL4 driver line targeting the tergotrochanteral muscle motor neuron (TTMn) |
| Genetic reagent (*Drosophila melanogaster*) | VT049105 | doi: 10.1101/198648 | flweb.janelia.org:VT049105 | split-GAL4 driver line targeting the peripherally synapsing interneuron (PSI) and other VNC interneurons |
| Genetic reagent (*Drosophila melanogaster*) | VT007170-x-VT041658 | this paper | split_gal4.janelia.org:SS02628 | split-GAL4 driver line targeting an interneuron |

*Continued on next page*

*Continued*

| Reagent type (species) or resource | Designation | Source or reference | Identifiers | Additional information |
|---|---|---|---|---|
| Genetic reagent (*Drosophila melanogaster*) | R26E02-x-R81D05 | this paper | split_gal4.janelia.org:SS01592 | split-GAL4 driver line targeting DNp18, DNp22, and the motor neuron for wing steering muscle III1 (III1mn) |
| Genetic reagent (*Drosophila melanogaster*) | R31B08-x-R24A03 | this paper | split_gal4.janelia.org:SS00724 | split-GAL4 driver line targeting DNp03, DNp28, DNp02 |
| Genetic reagent (*Drosophila melanogaster*) | R25C08-x-R79H02 | this paper | split_gal4.janelia.org:SS00728 | split-GAL4 driver line targeting DNp01 |
| Genetic reagent (*Drosophila melanogaster*) | R58E07-x-R39H12 | this paper | split_gal4.janelia.org:SS00733 | split-GAL4 driver line targeting DNp31 |
| Genetic reagent (*Drosophila melanogaster*) | R73C04-x-R39H12 | this paper | split_gal4.janelia.org:SS00734 | split-GAL4 driver line targeting DNb05 |
| Genetic reagent (*Drosophila melanogaster*) | R21A07-x-R72A01 | this paper | split_gal4.janelia.org:SS00736 | split-GAL4 driver line targeting DNa05 |
| Genetic reagent (*Drosophila melanogaster*) | R81D05-x-R81E05 | this paper | split_gal4.janelia.org:SS00738 | split-GAL4 driver line targeting DNp03, DNp18, DNp20, DNp22 |
| Genetic reagent (*Drosophila melanogaster*) | VT040541-x-VT050661 | this paper | split_gal4.janelia.org:SS01048 | split-GAL4 driver line targeting DNd01 |
| Genetic reagent (*Drosophila melanogaster*) | R21C05-x-R28E01 | this paper | split_gal4.janelia.org:SS01058 | split-GAL4 driver line targeting DNg09 |
| Genetic reagent (*Drosophila melanogaster*) | R30C01-x-R85H01 | this paper | split_gal4.janelia.org:SS01065 | split-GAL4 driver line targeting DNp04 |
| Genetic reagent (*Drosophila melanogaster*) | R31B08-x-R88C07 | this paper | split_gal4.janelia.org:SS01067 | split-GAL4 driver line targeting DNp03 |
| Genetic reagent (*Drosophila melanogaster*) | R32C05-x-R70C05 | this paper | split_gal4.janelia.org:SS01068 | split-GAL4 driver line targeting DNd02, DNd03 |
| Genetic reagent (*Drosophila melanogaster*) | R38H06-x-R65D06 | this paper | split_gal4.janelia.org:SS01070 | split-GAL4 driver line targeting DNa10 |
| Genetic reagent (*Drosophila melanogaster*) | R40H12-x-R85H01 | this paper | split_gal4.janelia.org:SS01071 | split-GAL4 driver line targeting DNp04, DNp20 |
| Genetic reagent (*Drosophila melanogaster*) | R42B02-x-R65C02 | this paper | split_gal4.janelia.org:SS01072 | split-GAL4 driver line targeting DNg07 |
| Genetic reagent (*Drosophila melanogaster*) | R50D07-x-R74C01 | this paper | split_gal4.janelia.org:SS01076 | split-GAL4 driver line targeting DNp04 |
| Genetic reagent (*Drosophila melanogaster*) | VT059450-x-VT037825 | this paper | split_gal4.janelia.org:SS01551 | split-GAL4 driver line targeting DNg01 |
| Genetic reagent (*Drosophila melanogaster*) | R10A07-x-VT033947 | this paper | split_gal4.janelia.org:SS01555 | split-GAL4 driver line targeting DNp27 |
| Genetic reagent (*Drosophila melanogaster*) | R91C05-x-R10A12 | this paper | split_gal4.janelia.org:SS01568 | split-GAL4 driver line targeting DNp20 |
| Genetic reagent (*Drosophila melanogaster*) | VT045148-x-VT045150 | this paper | split_gal4.janelia.org:SS01574 | split-GAL4 driver line targeting DNg28 |
| Genetic reagent (*Drosophila melanogaster*) | VT004985-x-VT017928 | this paper | split_gal4.janelia.org:SS01583 | split-GAL4 driver line targeting DNp11, DNp28 |
| Genetic reagent (*Drosophila melanogaster*) | VT034258-x-R32B03 | this paper | split_gal4.janelia.org:SS01584 | split-GAL4 driver line targeting DNg15 |
| Genetic reagent (*Drosophila melanogaster*) | R10A07-x-R24D07 | this paper | split_gal4.janelia.org:SS01585 | split-GAL4 driver line targeting DNp27 |
| Genetic reagent (*Drosophila melanogaster*) | R29F12-x-R24A03 | this paper | split_gal4.janelia.org:SS01594 | split-GAL4 driver line targeting DNp02, DNp03, DNp28 |
| Genetic reagent (*Drosophila melanogaster*) | R29F12-x-R31B08 | this paper | split_gal4.janelia.org:SS01595 | split-GAL4 driver line targeting DNp03 |

*Continued on next page*

*Continued*

| Reagent type (species) or resource | Designation | Source or reference | Identifiers | Additional information |
|---|---|---|---|---|
| Genetic reagent (*Drosophila melanogaster*) | R42B02-x-VT015782 | this paper | split_gal4.janelia.org:SS01598 | split-GAL4 driver line targeting DNg06 |
| Genetic reagent (*Drosophila melanogaster*) | R47D05-x-R39H12 | this paper | split_gal4.janelia.org:SS01599 | split-GAL4 driver line targeting DNa04 |
| Genetic reagent (*Drosophila melanogaster*) | R59B10-x-R21H11 | this paper | split_gal4.janelia.org:SS01601 | split-GAL4 driver line targeting DNb02 |
| Genetic reagent (*Drosophila melanogaster*) | R72H09-x-VT037825 | this paper | split_gal4.janelia.org:SS01603 | split-GAL4 driver line targeting DNg01 |
| Genetic reagent (*Drosophila melanogaster*) | R77H03-x-R74B04 | this paper | split_gal4.janelia.org:SS01605 | split-GAL4 driver line targeting DNb01 |
| Genetic reagent (*Drosophila melanogaster*) | VT015782-x-VT061933 | this paper | split_gal4.janelia.org:SS02260 | split-GAL4 driver line targeting DNg06 |
| Genetic reagent (*Drosophila melanogaster*) | VT020379-x-VT043626 | this paper | split_gal4.janelia.org:SS02541 | split-GAL4 driver line targeting DNg01 |
| Genetic reagent (*Drosophila melanogaster*) | R22C05-x-VT025718 | this paper | split_gal4.janelia.org:SS02545 | split-GAL4 driver line targeting DNa02 |
| Genetic reagent (*Drosophila melanogaster*) | R81C11-x-VT025598 | this paper | split_gal4.janelia.org:SS02555 | split-GAL4 driver line targeting DNg11 |
| Genetic reagent (*Drosophila melanogaster*) | VT029814-x-VT008145 | this paper | split_gal4.janelia.org:SS02610 | split-GAL4 driver line targeting DNp13 |
| Genetic reagent (*Drosophila melanogaster*) | R58E07-x-R30C01 | this paper | split_gal4.janelia.org:SS02621 | split-GAL4 driver line targeting DNp31 |
| Genetic reagent (*Drosophila melanogaster*) | R38H06-x-R50B07 | this paper | split_gal4.janelia.org:SS04160 | split-GAL4 driver line targeting DNg14 |
| Genetic reagent (*Drosophila melanogaster*) | R40F04-x-R83B06 | this paper | split_gal4.janelia.org:SS04530 | split-GAL4 driver line targeting DNc01, DNc02 |
| Genetic reagent (*Drosophila melanogaster*) | VT042835-x-R65C10 | this paper | split_gal4.janelia.org:SS05089 | split-GAL4 driver line targeting DNg02 |
| Genetic reagent (*Drosophila melanogaster*) | R44A07-x-VT062604 | this paper | split_gal4.janelia.org:SS05099 | split-GAL4 driver line targeting DNg28 |
| Genetic reagent (*Drosophila melanogaster*) | VT023750-x-VT042835 | this paper | split_gal4.janelia.org:SS05107 | split-GAL4 driver line targeting DNg02 |
| Genetic reagent (*Drosophila melanogaster*) | VT042835-x-VT023750 | this paper | split_gal4.janelia.org:SS05116 | split-GAL4 driver line targeting DNg02 |
| Genetic reagent (*Drosophila melanogaster*) | VT042835-x-VT020379 | this paper | split_gal4.janelia.org:SS05122 | split-GAL4 driver line targeting DNg01 |
| Genetic reagent (*Drosophila melanogaster*) | pJFRC51-3xUAS-Syt::smGFP-HA in su(Hw)attPa | *Aso et al., 2014* | | |
| Genetic reagent (*Drosophila melanogaster*) | pJFRC22-10XUAS-IVS-myr::tdTomato | *Pfeiffer et al. 2010* | https://www.janelia.org/lab/rubin-lab/tools-reagents | |
| Genetic reagent (*Drosophila melanogaster*) | pJFRC93-13XLexAop2-IVS-Syn21-mPA-p10 (VK00005) | *Pfeiffer et al. 2012* | https://www.janelia.org/lab/rubin-lab/tools-reagents | |
| Genetic reagent (*Drosophila melanogaster*) | nsyb-LexAp65 (attP2)/ TM2 | Rubin, G. | | Personal communicaiton |
| Genetic reagent (*Drosophila melanogaster*) | mPA-LexAop | *Pfeiffer et al., 2012* | | |
| Genetic reagent (*Drosophila melanogaster*) | tsh-LexA | *Simpson, 2016* | | |
| Genetic reagent (*Drosophila melanogaster*) | LexAop2-GAL80 | Pfeiffer, B. | http://flybase.org/reports/FBrf0212441 | |

*Continued on next page*

*Continued*

| Reagent type (species) or resource | Designation | Source or reference | Identifiers | Additional information |
|---|---|---|---|---|
| Genetic reagent (*Drosophila melanogaster*) | pJFRC20-8XLexAop2-IVS-GAL80-WPRE (su(Hw)attP5) | *Pfeiffer et al., 2010* | https://www.janelia.org/lab/rubin-lab/tools-reagents | |
| Genetic reagent (*Drosophila melanogaster*) | pJFRC28-10XUAS-IVS-GFP-p10(attP40) | *von Reyn et al., 2014* | | |
| Genetic reagent (*Drosophila melanogaster*) | pJFRC200-10XUASIVS-myr::smGFP-HA in attP18 | *Aso et al., 2014* | | |
| Genetic reagent (*Drosophila melanogaster*) | pJFRC51-3xUAS | *Aso et al., 2014* | | |
| Genetic reagent (*Drosophila melanogaster*) | Syt::smGFP-HA in su(Hw) attP1 | *Nern et al., 2015* | | |
| Genetic reagent (*Drosophila melanogaster*) | pJFRC225-5xUAS-IVS-myr::smGFP-FLAG in VK00005 | *Nern et al., 2015* | http://flybase.org/reports/FBrf0228639.html | |
| Genetic reagent (*Drosophila melanogaster*) | 20XUAS-CsChrimson-mVenus trafficked in attP18 | *Klapoetke et al., 2014* | http://flybase.org/reports/FBrf0224686 | |
| Genetic reagent (*Drosophila melanogaster*) | teashirt-GAL80 | *Rubinstein et al., 2010* | | |
| Antibody | nc82 supernatant | Mouse α-bruchpilot. | Developmental Studies Hybridoma Bank. # nc82-s | https://www.janelia.org/sites/default/files/Project%20Teams/Fly%20Light/FL%20Protocol%20-%20Adult%20IHC%20-%20Split%20Screen_1.pdf |
| Antibody | rabbit polyclonal anti-GFP | Thermo Fisher Scientific | Cat #: A-11122; RRID: AB_221569 | |
| Antibody | Alexa Fluor 488 goat anti-rabbit | Thermo Fisher Scientific | Cat #: A-11034; RRID: AB_2576217 | |
| Antibody | Alexa Fluor 568 goat anti-mouse | Thermo Fisher Scientific | Cat #: A-11031; RRID: AB_144696 | |
| Chemical compound, drug | paraformaldehyde | Electron Microscopy Sciences | 15713-S | https://www.janelia.org/sites/default/files/Project%20Teams/Fly%20Light/FL%20Recipe%20-%20PFA_2.pdf |
| Chemical compound, drug | Triton X-100 | Sigma Aldrich | X100 | https://www.janelia.org/sites/default/files/Project%20Teams/Fly%20Light/FL%20Protocol%20-%20Adult%20IHC%20-%20Split%20Screen_1.pdf |
| Chemical compound, drug | DPX Mountant | Electron Microscopy Sciences | #13512 | https://www.janelia.org/sites/default/files/Project%20Teams/Fly%20Light/FL%20Protocol%20-%20DPX%20Mounting_0.pdf |
| Chemical compound, drug | xylene | Fisher Scientific | x5-500 | https://www.janelia.org/sites/default/files/Project%20Teams/Fly%20Light/FL%20Protocol%20-%20DPX%20Mounting_0.pdf |

## Mass-staining of DNs with photoactivatable GFP

To quantify the number of neurons with cell bodies in the brain and axonal projections into the VNC, we pan-neuronally expressed phtotoactivatable GFP (PA-GFP) using the following fly stock: w1118; pJFRC22-10XUAS-IVS-myr::tdTomato (attP40)/CyO; pJFRC93-13XLexAop2-IVS-Syn21-mPA-p10 (VK00005), nsyb-LexAp65 (attP2)/TM2. This line drives the PA-GFP construct, LexAop2-mPA (*Pfeiffer et al., 2012*; *Patterson and Lippincott-Schwartz, 2002*), under control of the *neuronal*

*synaptobrevin* promotor, *nsyb*-LexA (also known as R57C10-LexA; *Jenett et al., 2012*), which codes for a synaptic vesicle protein that should be present in most neurons. Photoactivation was performed on adult female progeny aged 0–1 days after eclosion. The central nervous system was dissected intact and kept in saline solution. Using a confocal microscope (Zeiss LSM 710), we scanned a volume of the neck connective posterior to the cervical nerve every 5 min for 1 hr with 408 nm light. After the activation, the brain was imaged, first from the anterior surface, and then from the posterior surface, by flipping the sample. Labeled cell bodies in the brain were counted manually in four preparations.

## Screening for DNs

To identify driver lines containing specific DN morphologies, we manually searched the brain and VNC expression pattern images of approximately 9000 GAL4 driver lines from publically available GAL4 lines in the Janelia FlyLight collection (http://flweb.janelia.org, *Jenett et al., 2012*) for neurons with an axon in the neck connective and a cell body in the brain. To make individual neuron morphology more clear, and to help distinguish DNs from ascending neurons, we used *teashirt (tsh)*, a transcription factor specifying the VNC (*Röder et al., 1992*; *Simpson, 2016*). We chose 586 GAL4 lines with connective expression and crossed these to a line with *tsh*-LexA, LexAop2-GAL80, and a GFP reporter (w; tsh-LexAp65, pJFRC20-8XLexAop2-IVS-GAL80-WPRE (su(Hw)attP5), pJFRC28-10XUAS-IVS-GFP-p10 (attP40)/CyO, TB-RFP; *Simpson, 2016*; *Pfeiffer et al., 2010*). For the progeny, this suppressed expression in most neurons originating in the VNC, allowing us to clearly visualize the morphologies of any DN axon terminals (see Expression pattern visualization below). We generated approximately 200 new transgenic lines using enhancers identified in our screen to express either the GAL4 transcription activation domain (p65ADZp) or the DNA binding domain (ZpGAL4DBD). New lines were made using vectors from *Pfeiffer et al. (2010)*.

## Expression pattern visualization

Expression patterns of GAL4 driver lines crossed with *tsh*-LexA were obtained by CNS dissection, immunohistochemistry, and confocal imaging. We used the standard Janelia FlyLight protocols (based on *Jenett et al., 2012*), available at https://www.janelia.org/project-team/flylight/protocols, which we describe briefly here. The complete central nervous systems of 3- to 5-day-old female adult progeny were dissected in S2 media (Schneider's Insect Medium, Sigma), fixed in paraformaldehyde, and transferred to a goat serum blocking buffer for 1 hr. The buffer was then replaced with the primary antibodies (mouse nc82 supernatant at 1:30, rabbit polyclonal anti-GFP at 1:1000) diluted in phosphate buffered saline with 0.5% Triton X-100 (PBT) and rocked at 4°C for 36–48 hr. After washing with PBT, the samples were next incubated with secondary antibodies (Alexa Fluor 488 goat anti-rabbit, and Alexa Fluor 568 goat anti-mouse at 1:400) diluted in PBT and rocked at 4°C for 3 days. Next, samples were washed, fixed again in paraformaldehyde, mounted on a poly-L-lysine cover slip, cleared with xylene, and embedded in dibutyl phthalate in xylene (DPX) on a standard microscope slide with spacers. After drying for two days, samples were imaged at 20X with a confocal microscope (Zeiss LSM 510) (*Dionne et al., 2018*).

## Split-GAL4 intersections

Based on our screening of GAL4 and GAL4 with *teashirt* lines, we selected AD/DBD combinations from the Janelia (*Dionne et al., 2018*) and VT (*Tirian and Dickson, 2017*) collections (that we thought shared expression in individual DNs. To visualize combined expression patterns, we crossed males carrying a GFP reporter (pJFRC200-10XUASIVS-myr::smGFP-HA in attP18) and the ZpGAL4DBD transgene (in attP2) with virgin females carrying the p65ADZp transgene in either su(Hw)attP8, attP40, or VK00027 and examined expression in 3- to 10-day-old female progeny as described above. The split-GAL4 combinations that we deemed sparse enough to include in our DN collection were made into stable stocks containing the AD and DBD transgenes. To obtain polarity and higher resolution (40x, 63x) information on selected lines, split-GAL4 lines were crossed to pJFRC51-3xUAS-Syt::smGFP-HA in su(Hw)attP1; pJFRC225-5xUAS-IVS-myr::smGFP-FLAG in VK00005 and processed for imaging. We used the multicolor flip out technique to stochastically label individual neurons in lines that contained multiple cells (*Nern et al., 2015*). These protocols are available on the Janelia FlyLight website (https://www.janelia.org/project-team/flylight/protocols).

Some split-GAL4 lines were also crossed to 20XUAS-CsChrimson-mVenus trafficked in attP18 (virginator stock) and processed as above to visualize expression pattern when using the CsChrimson effector, as observed expression patterns are known to vary slightly depending on the reporter used (*Aso et al., 2014*). Based on their GFP or CsChrimson expression patterns, we made our best estimate of the number of background (non-targeted-DN) cell types in each split-GAL4 line made, and we gave each split line a quality score of A (no background expression), B (one background cell type), or C (two or more background cell types). Confocal image stacks of the stabilized split-GAL4 intersections are available online (http://www.janelia.org/split-gal4).

### Terminology and neuropil annotation

We operationally define a DN as a neuron that: (1) connects brain and VNC with an axon through the neck connective and (2) has its cell body in the brain. We did not include ascending neurons and neck motor neurons in this analysis. For the physical definitions and nomenclature of the brain neuropils, we followed the virtual fly brain (http://www.virtualflybrain.org/; *Ito et al., 2014*). Abbreviations used in the document are summarized in *Supplementary file 6*. For the terminology of the VNC, we primarily followed *Power (1948)*, *Merritt and Murphey (1992)* and *Boerner and Duch, 2010* and introduced some new formal standards also described in *Court et al. (2017)*. DNs are named with first two letters 'DN' followed by a single letter indicating the location of the cell body (a, anterior dorsal; b, anterior ventral; c, pars intercerebralis; d, outside cluster on anterior surface; g, gnathal ganglion; p, posterior surface; x, outside brain) and then a two-digit number assigned randomly (roughly in discovery order) within each cell body group.

### Image processing

Neuron and neuropil tracing was carried out semi-manually using Amira 5.4.3 (Visage Imaging, Fuerth, Germany). For the reconstruction of neuropils, individual objects were smoothed using the 'smooth label' function after the manual tracing. Volume rendering was performed using Amira 'generate surface' function. Segmentation data for the brain in some figures (*Figures 6* and *9*) were obtained from Virtual Fly Brain (https://github.com/VirtualFlyBrain/DrosAdultBRAINdomains). Segmentation data for the neuronal tracts in the VNC was kindly provided by Jana Boerner (*Boerner and Duch, 2010*). For images of single DN morphology, masked images were used for visualization. We performed segmentation for individual neurons in the confocal stacks of generation-1 GAL4 lines with sparse expression pattern or split-GAL4 lines. We first detected the signal with the Amira 'Interactive Thresholding' function. We then corrected any false detection by manual tracing. Using this image as a mask, we obtained the final masked images shown in the figures using a custom-made program written in MATLAB and the image processing toolbox (MathWorks, Natick, MA, USA). The contrast and brightness of images were modified in Image J (National Institutes of Health, Bethesda, MD). Figures were prepared in Adobe Illustrator CS (Adobe systems, San Jose, CA).

Confocal image stacks of split-GAL4 expression patterns in the brain were aligned to standardized brain template JFRC2013 (available here: https://github.com/jefferislab/BridgingRegistrations) as part of an established processing pipeline (*Aso et al., 2014*; *Peng et al., 2011*). A similar template was derived from the nc82 expression pattern in the VNC of an example female CantonS fly imaged by the FlyLight Project team (template is available here: https://github.com/VirtualFlyBrain/DrosAdultVNSdomains/blob/master/template/Neuropil_185.nrrd). Our VNC alignment pipeline was adapted from *Court et al. (2017)*. Briefly: confocal VNC stacks were first converted to an 8-bit nrrd file format, preprocessed using the nc82 reference channel to normalize contrast across samples, rotated to approximately orient the VNC along the anterior-posterior axis, and then the reference channel was aligned to the template by nonrigid warping (*Rohlfing and Maurer, 2003*) using the Computational Morphometry Toolkit (https://www.nitrc.org/projects/cmtk/), as described in detail in *Jefferis et al. (2007)*. The signal channel containing the GFP expression pattern was then transformed using the warped mesh determined above, and the two individual image files were combined as separate channels and converted back to the LSM file format.

## Analysis

We performed hierarchical clustering based on the neurite innervation of DNs in the brain (*Figure 8*) and VNC (*Figure 7*) using the MATLAB statistics toolbox with Pearson's correlation as metric and average linkage for calculating distances. DN innervation was represented as a matrix, where the presence and absence of innervation were scored as '1' and '0' (*Supplementary file 1*). We only scored smooth processes as '1' for DN innervation in the brain. To evaluate the pattern of DN axonal projections in the VNC, we calculated Pearson's linear correlation coefficient between each pair of VNC regions (*Figure 7B*). To evaluate the pattern of DN axonal projections in the brain, we calculated Pearson's linear correlation coefficient between each pair of brain regions and sorted by the results of the clustering analysis (*Figure 8B*). The brain regions without DN innervation were omitted.

For the visualization of spatial patterns of DN distribution (*Figures 6* and *9*), we made pseudo-color maps for the brain and VNC neuropil compartments. The number of DN types visiting each compartment was counted and mapped onto the neuropil segmentation data with pseudo-color. We only counted the innervation with smooth appearance in *Figure 6A*, *top*, and 9A, and the innervation with varicose appearance in *Figure 6A*, *bottom*, and 9B.

For the analysis of neurite distribution in the protoerebrum (*Figure 15*), registered brain data were used to compare the dendritic distribution of DNs in the protocerebrum (*Peng et al., 2011*). Using segmentation data, we counted the number of voxels for neurite volume along the anterior-posterior axis with 1 μm interval. We excluded the volume of somata and axons. The value at each depth was normalized by the maximum value for each DN.

## Acknowledgements

We are grateful to Gerald Rubin and Barry Dickson for providing GAL4 lines, Barett Pfeiffer and Gerry Rubin for constructs used to make the PA-GFP reagent, Julie Simpson, Heather Dionne and Teri Ngo for genetic reagents, Rob Court for initial help setting up the VNC aligner at Janelia and Hideo Otsuna for optimization of the VNC aligner and processing of VNC data, Jana Boerner for segmentation data of VNC bundles. The Janelia Fly facility (Amanda Cavallaro, Todd Laverty, Karen Hibbard, Jui-Chun Kao and others) helped in fly husbandry, and the FlyLight Project Team (https://www.janelia.org/project-team/flylight, Rebecca Johnston, Oz Malkesman, Nirmala Iyer, Kevin Zeng, Kelley Salvesen, Nick Abel, Phuson Hulamm, Reeham Motaher, Susana Tae, Rebecca Vorimo) performed brain dissections, histological preparations, and confocal imaging. We also thank Kei Ito and Masayoshi Ito for sharing information on clonal units, Jens Goldammer and Masayoshi Ito for comments on an early version of the manuscript, and Jim Truman and David Shepherd for helpful discussions on VNC anatomy. We are grateful to the FlyCircuit database from the NCHC (National Center for High-performance Computing) and NTHU (National Tsing Hua University), and to the FLYBRAIN neuron database in the University of Tokyo. This research was partially funded by the Descending Interneuron Project Team (https://www.janelia.org/project-team/fly-descending-interneuron) and the Visiting Scientist Program at The Janelia Research Campus.

## Additional information

### Funding

| Funder | Author |
| --- | --- |
| Howard Hughes Medical Institute | Gwyneth M Card |

The funders had no role in study design, data collection and interpretation, or the decision to submit the work for publication.

### Author contributions

Shigehiro Namiki, Conceptualization, Data curation, Software, Formal analysis, Investigation, Visualization, Methodology, Writing—original draft, Writing—review and editing; Michael H Dickinson, Conceptualization, Visualization, Methodology, Writing—original draft, Writing—review and editing;

Allan M Wong, Investigation, Methodology, Data acquisition; Wyatt Korff, Conceptualization, Data curation, Funding acquisition, Visualization, Methodology, Writing—original draft, Project administration, Writing—review and editing; Gwyneth M Card, Conceptualization, Data curation, Supervision, Funding acquisition, Visualization, Methodology, Writing—original draft, Project administration, Writing—review and editing

**Author ORCIDs**
Shigehiro Namiki http://orcid.org/0000-0003-1559-799X
Michael H Dickinson http://orcid.org/0000-0002-8587-9936
Allan M Wong http://orcid.org/0000-0002-8492-2162
Wyatt Korff http://orcid.org/0000-0001-8396-1533
Gwyneth M Card http://orcid.org/0000-0002-7679-3639

**Decision letter and Author response**
Decision letter https://doi.org/10.7554/eLife.34272.062
Author response https://doi.org/10.7554/eLife.34272.063

## Additional files

**Supplementary files**
• Supplementary file 1. Morphological characteristics of identified descending neurons. Column 1: DN cell type; column 2: maximum observed number of neurons in a cell type from a single preparation; column 3: maximum cell count for that type in a single sample; column 4: example Gen-1 GAL4 or split-GAL4 line with expression in the cell type; column 5: Y, yes, or N, no whether the cell type contains projections across the midline; column 6: indicates whether the axon traverses the cervical connective ipsilateral (I) or contralateral (C) to the cell body; columns under 'Brain,' 'VNC,' 'Tract': number indicates whether the cell type has neurites in the specified neuropil that are smooth (1), varicose (2), or both (3); 0 indicates the cell type has no innervation to the specified compartment; 'Tract' refers to the VNC tract traversed by the axon.
DOI: https://doi.org/10.7554/eLife.34272.054

• Supplementary file 2. Descending neurons with similar morphology reported in *Drosophila melanogaster* and other species.
DOI: https://doi.org/10.7554/eLife.34272.055

• Supplementary file 3. DN Split-GAL4 lines generated and used in this study. 'Line name' refers to the split-GAL4 stock number by which the line can be ordered. 'Genotype stock name' indicates which Gen1 parent lines were crossed to create the split line. 'DN type' indicates which DN cell type is targeted by the line. 'Sample' is the number of samples for which driver line expression was visualized (see Materials and methods). 'Stochastic pattern' is the number of samples that showed an expression patter deviant from the most common expression pattern observed. 'bg' indicates the approximate number of off-target cell types that also have expression using the driver line.
DOI: https://doi.org/10.7554/eLife.34272.056

• Supplementary file 4. Interneuron and motor neuron split-GAL4 lines generated in this study.
DOI: https://doi.org/10.7554/eLife.34272.057

• Supplementary file 5. More broadly expressing DN split-GAL4 lines made in this study. These lines were not used in the present analysis. A rough quality score is provided for each line indicating it sparseness: Quality A, sparse lines without background expression; Quality B, lines with limited background expression or <5 off-target cell types; Quality C, lines with background expression.
DOI: https://doi.org/10.7554/eLife.34272.058

• Supplementary file 6. Abbreviations used.
DOI: https://doi.org/10.7554/eLife.34272.059

• Transparent reporting form
DOI: https://doi.org/10.7554/eLife.34272.060

## Data availability

Confocal images of driver line expression data can be found at http://splitgal4.janelia.org/cgi-bin/splitgal4.cgi. Source data are provided in editable tables in. xlsx format.

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
