## [Decision Letter]

Thank you for submitting your article "The functional organization of descending sensory-motor pathways in *Drosophila*" for consideration by *eLife*. Your article has been reviewed by 3 peer reviewers, and the evaluation has been overseen by Kristin Scott as the Reviewing Editor (who served as one of the reviewers) and Gary Westbrook as the Senior Editor. The following individual involved in review of your submission has agreed to reveal his identity: Andrew D Straw. The reviewers have discussed the reviews with one another and the Reviewing Editor has drafted this decision to help you prepare a revised submission.

Summary:

In this manuscript, Card and colleagues present a thorough anatomical characterization of half of the descending neurons of the adult (*Drosophila*) fly. They reveal the existence of three main descending pathways. Although the very nature of this work could have yielded a dry and highly technical study, the present manuscript is just the opposite: it is a read that will intellectually engage any neuroscientist, anatomist or not. The functional implications of the mapping of the descending pathways are discussed based on numerous references to published literature in *Drosophila* and in other insects. The Discussion section of the paper represents a massive effort to combine new and previously published observations in a framework that permits the generation of functional hypothesis. Technically, this manuscript should also have a long-lasting impact by introducing a solid methodology to map and compare patterns of pre- and post-synaptic regions of neurons with related functions. The care with which the anatomical patterns of the descending neurons is documented and compared sets an example to be followed in future work. Finally, the quality and the content of the supporting material is spectacular. Many supplementary figures provide a wealth of information that the reader will find extremely valuable (the authors should be praised for this sharing of extra information). In summary, this manuscript stands as a monumental achievement that will empower the study of sensorimotor integration in *Drosophila*.

Essential revisions:

1) Given that many readers will use this manuscript to identify split-Gal4 lines suited to monitor and manipulate their favorite descending neuron, it would be important to comment on the reproducibility of the expression pattern of a given split-Gal4 lines across animals. Did the authors observe significant variability in the regions covered by the dendrites and axon terminals of the same split-Gal4 line? Although the reliability of the expression patterns of split-Gal4 lines should not impact the conclusions of the work, it will influence the future use of the Gal4 lines for targeted loss-of-function and gain-of-function manipulations. Can the reader assume that the expression of a given split-Gal4 line will be consistent across individuals? Could they authors establish this result for a few representative lines? On a related note, did the authors observe any split-Gal4 line(s) associated with a unilateral labeling of descending neurons?

2) The authors go to great lengths to catalogue the pre- and post-synaptic innervation patterns of each class of descending neuron. Based on these individual patterns, a statistical analysis of connectivity features between regions of the brain and the VNC is reported in Figure 9. Could this analysis be pushed further by predicting the projection of the descending neuron to a given region of the VNC based on the knowledge of its dendritic patterns? Are there any combinations of brain regions that are highly predictive of specific features associated with the projection in the VNC? Is there a one-to-one relationship or many-to-one between the innervation patterns in the brain and the VCN?

3) A supplemental table that is easily searchable would be useful; i.e., a table with DN name, approximate #of other neurons labeled, DN Input neuropil, DN output neuropil, DN flycircuit ID if available so that one can NBLAST.

---

## [Author Response]

Essential revisions:

1) Given that many readers will use this manuscript to identify split-Gal4 lines suited to monitor and manipulate their favorite descending neuron, it would be important to comment on the reproducibility of the expression pattern of a given split-Gal4 lines across animals. Did the authors observe significant variability in the regions covered by the dendrites and axon terminals of the same split-Gal4 line? Although the reliability of the expression patterns of split-Gal4 lines should not impact the conclusions of the work, it will influence the future use of the Gal4 lines for targeted loss-of-function and gain-of-function manipulations. Can the reader assume that the expression of a given split-Gal4 line will be consistent across individuals? Could they authors establish this result for a few representative lines? On a related note, did the authors observe any split-Gal4 line(s) associated with a unilateral labeling of descending neurons?

We agree it would help potential users of the lines to provide some information about stochastic expression. For each split-GAL4 line stabilized, we checked expression patterns in an average of 4 brains. Most lines showed stable expression across all samples. We found 19/132 (14%) of lines had stochastic expression in one or more samples. This includes 8 lines for which the stochastic expression resulted in unilateral labeling. We have updated Table 3 to include our observations of stochastic expression for almost every line, and we have added the following text to the Results section:

“Drosophila transgenes, including those used to create split-GAL4 driver lines, may express stochastically (Pfeiffer et al., 2010), Aso et al., 2014). […] We note that expression patterns may also be affected by transgene location and properties of the reporter to which the split-GAL4 driver lines are crossed and thus should be verified for any particular experiment.”

2) The authors go to great lengths to catalogue the pre- and post-synaptic innervation patterns of each class of descending neuron. Based on these individual patterns, a statistical analysis of connectivity features between regions of the brain and the VNC is reported in Figure 9. Could this analysis be pushed further by predicting the projection of the descending neuron to a given region of the VNC based on the knowledge of its dendritic patterns? Are there any combinations of brain regions that are highly predictive of specific features associated with the projection in the VNC? Is there a one-to-one relationship or many-to-one between the innervation patterns in the brain and the VCN?

We agree with the reviewers that these are interesting questions. We have already analyzed the clustering of DN dendritic patterns in Figure 8A. A straightforward way to visualize what the reviewers are asking for is to take this dendritic pattern clustering and see whether it produces any clustering of the VNC outputs. We now provide this visualization as Figure 9—figure supplement 2. We have also added the complementary analysis – i.e. try to predict patterns of dendritic input from the patterns of VNC innervation (Figure 9—figure supplement 1). We think that this analysis bears out our observation of three prominent brain-VNC pathways and is sufficient to say that, with the exception of a selective connection between LC4 and the lower tectulum, there is generally no one-to-one relationship between the brain and VNC regions, even on a patterned basis.

In reference to these new figures, we have added the following text:

“An analysis visualizing brain innervation patterns based on VNC clusters (Figure 9—figure supplement 1), and vice versa (Figure 9—figure supplement 2), indicates that the three main pathways identified here are the most prominent organization to emerge from the current analysis.”

Besides this additional analysis, we also make available an Excel spreadsheet version of Supplementary file 1, which is the annotated dendritic and VNC output pattern for each DN catalogued. We hope others will use this data to further explore the input/output relationships represented by this population.

3) A supplemental table that is easily searchable would be useful; i.e., a table with DN name, approximate #of other neurons labeled, DN Input neuropil, DN output neuropil, DN flycircuit ID if available so that one can NBLAST.

We will provide Excel spreadsheet versions of Supplementary files 1 and 3. We have expanded Supplementary file 3 to include approximate number of other neurons labeled.